



# The behaviour of charged particles (ions) during new particle formation events in urban Leipzig (Germany)

Alex Rowell[1], James Brean[1], David C.S. Beddows[1], Zongbo Shi[1], Avinash Kumar[2], Matti Rissanen[2,3], Miikka Dal Maso[4], Peter Mettke[5], Kay Weinhold[5], Maik Merkel[5], Roy M. Harrison[1,6]

[1]School of Geography, Earth & Environmental Sciences, University of Birmingham, Birmingham B15 2TT, United Kingdom
[2]Aerosol Physics Laboratory, Physics Unit, Tampere University, 33720 Tampere, Finland
[3]Department of Chemistry, University of Helsinki, P. O. Box 55, Helsinki, Finland
[4]Department of Physics, Tampere University of Technology, P.O. Box 692, 33100 Tampere, Finland
[5]Leibniz Institute for Tropospheric Research (TROPOS), Atmospheric Chemistry Department (ACD), Permoserstr. 15, 04318 Leipzig, Germany
[6]Department of Environmental Sciences, Faculty of Meteorology, Environment and Arid Land
Agriculture, King Abdulaziz University, Jeddah 21589, Saudi Arabia

*Correspondence to*: Roy M. Harrison (r.m.harrison@bham.ac.uk)

## ABSTRACT

Air ions are electrically charged molecules or particles in air. They are found in the natural
environment. Charging accelerates the formation and growth of new aerosol particles. A Neutral cluster and Air Ion Spectrometer was deployed in Leipzig, Germany, to measure the number size distribution of ions from 0.8 to 42 nm, between July 27[th] and August 25[th] 2022. Charged particles were mobility classified into small (0.8–1.6 nm), intermediate (1.6–7.5 nm), and large (7.5–22 nm) fractions and their mean concentrations (sum of positive and negative
polarities) during the campaign were 462, 88, and 420 cm[-3], respectively. The study found that small charged particles were primarily associated with radioactive decay during the early hours, while the intermediate and large charged fractions were linked to photochemistry and local air pollution, as indicated via synchronous peaks in sulphuric acid dimer and black carbon concentrations, respectively. NPF events, observed on 30% of days, coincided with intense solar
radiation. Small charged particle concentrations were lower on NPF event days, whereas the intermediate and large charged species exhibited higher concentrations. The apparent contributions of charged species to 3 and 7.5 nm particles formation rates were 5.7 and 12.7%, respectively, with mean growth rates of 4.0 and 5.2 nm h[-1]. Although the ratio of apparent formation rates for charged to uncharged nanoparticles of 3 nm suggested a minor role for charged





species in NPF, a substantial increase in intermediate and large charged species was associated with NPF events. The findings contribute valuable insights into the complex interplay between charged species and particle formation in urban environments.

## 1. INTRODUCTION

Atmospheric aerosol particles influence the Earth's energy budget (Carslaw et al., 2013; Quaas
et al., 2009), impair visibility (e.g. haze events, aerosol–fog interactions, and cloud formation) (Boutle et al., 2018; Tian et al., 2016), and adversely impact human health through the degradation of air quality (Kelly and Fussell, 2015). The environmental impacts and health effects of aerosol particles are dependent on their number concentration, size, structure, chemical composition, and charge state. These properties, however, vary spatially and temporally.


New particle formation (NPF) accounts for a large fraction of global aerosol production (Gordon et al., 2017; Spracklen et al., 2010). NPF is a phenomenon observed in many different environments around the world, from pristine remote locations to polluted urban atmospheres (Brean et al., 2021, 2023; Uusitalo et al., 2021; Yao et al., 2018). It is an important atmospheric
process wherein gas phase molecules cluster together and grow to form new aerosol particles. Air ions can play an important role in the enhancement of these formation and growth processes.

Air ions are electrically charged molecules or particles in the atmosphere which can influence
NPF processes. These can be positively or negatively charged, depending on whether a particle has gained or lost an electron. They can promote the formation of small molecular clusters, enhance their stability, and decrease their evaporation rate (He et al., 2021; Kirkby et al., 2011). Following nucleation and the formation of stable particles, charged particles persist as a source of charge, attracting molecules to particles and facilitating further particle growth (Svensmark
et al., 2017).

Various environmental factors impact the production and removal of air ions and charged particles in the atmosphere. Sources include cosmic rays (Svensmark et al., 2017), radioactive decay (Zhang et al., 2011), traffic (Jayaratne et al., 2014), transmissions lines (Jayaratne et al.,
2011), volcanic eruptions (Rose et al., 2019), thunderstorms and lightning (J-P Borra et al., 1997), solar radiation (Vana et al., 2008; Wang et al., 2005), vegetation (Wang and Li, 2009),



and splashing water (Tammet et al., 2009). Sinks involve redistribution via coagulation with pre–existing aerosol (Mahfouz and Donahue, 2021), losses via ion–ion recombination (Zauner-Wieczorek et al., 2022) and dry deposition (Tammet et al., 2006).


Several studies have investigated the role of ions in the nucleation process, yielding varied results. Manninen et al. (2010) found that contributions of ion–induced nucleation to total particle formation at 2 nm were typically in the range of 1–30% between 12 field sites across Europe. In other remote locations, Kulmala et al. (2010) found that contributions were typically

significantly less than 10% in Hyytiälä (Finland), Hohenpeissenberg (Germany), and Melpitz (Germany). In other urban locations, contributions were observed at approximately 1.3% at 1.5/2 nm in Helsinki, Finland (Gagné et al., 2012) and 10% at 3 nm in Brisbane, Australia (Pushpawela et al., 2018). However, few comprehensive analyses of the temporal variation of charged particles, together with their contribution to particle formation and growth in the urban

environment, have been published to date.

The aim of this paper was to better understand the behaviour of charged particles and their behaviour during atmospheric NPF in an urban environment. A Neutral cluster and Air Ion Spectrometer was deployed at an urban background site in Leipzig, Germany, to measure the

mobility distribution of neutral and charged particles, between 27[th] July and 25[th] August 2022. The urban background site is located in the Leibniz Institute for Tropospheric Research, a re-nowned centre specialising in both in–situ and remote observations of aerosols and clouds. Given its expertise and state–of–the–art facilities, the institute stood out as an ideal location for conducting a research campaign on charged species and their behaviour during NPF. The air

ion/charged particle population was mobility classified into small (0.8–1.6 nm), intermediate (1.6–7.5 nm), and large particles (7.5–22 nm) for analysis, following the classification system outlined by Tammet (2006).

## 2. MATERIALS AND METHODS

### 2.1. Site description

Leipzig is located in the German State of Saxony in east Germany. Leipzig is the 8[th] most populated city in Germany, with 0.6 million inhabitants. The measurements were located at the Leibniz Institute for Tropospheric Research e.V. (denoted as Leipzig–TROPOS) (N51°21'09", E12°26'04", 127 m above mean sea level) within the Leipzig Science Park (**Figure 1**), from





27th July to 25th August 2022. The charged and neutral particle measurements were taken from a laboratory on the fourth floor of an institute building positioned central to the Science Park. Leipzig–TROPOS is located in excess of 100 m from a number of highly–trafficked roads and is classified as an urban background site. The Science Park contains other research institutes and related companies, allotted parking bays, including a multi–storey carpark, and greenspace. The park perimeter includes transport infrastructure (including road, rail, and tramways), commercial property (e.g. restaurants, hotels, a petrol station etc.), residential property, on–street parking, and additional greenspace.

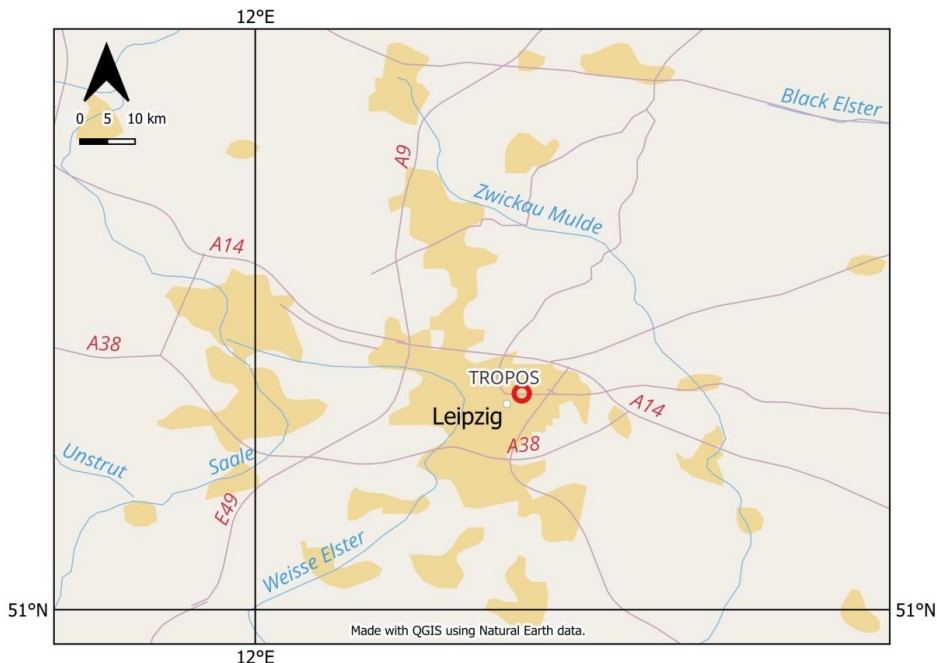

**Figure 1: Location of the TROPOS site (red marker), approximately 4 km northeast of Leipzig city centre. Made with QGIS using Natural Earth Data.**

**2.2. Meteorological conditions**

Leipzig has a temperate continental climate. The city's weather can be highly variable as it is exposed to both cold and warm air masses and thunderstorms are not uncommon during the warm season (May through August). Weather–related measurements were taken from a meteorological station on the roof of the same institute building accommodating the air quality–related instruments at Leipzig–TROPOS. From June to August 2022, persistent heatwaves affected many parts of Europe, including Germany. The mean hourly air temperature during the





campaign was 22.4 °C and the highest hourly air temperature was 37.1 °C recorded on the 4th August 2022.

### 2.3. Description of the instruments

#### 2.3.1. Neutral cluster and Air Ion Spectrometer

The principle of the Neutral cluster and Air Ion Spectrometer (NAIS, Ariel Ltd., Estonia) is described in detail by Mirme and Mirme (2013). An NAIS was used to measure the charged particle number size distribution (PNSD) from 0.8–42 nm, and the neutral PNSD from 3–42 nm by their mobilities (3.2 to 0.0013 $cm^2$ $V^{-1}$ $s^{-1}$) for the duration of the measurement campaign.

The instrument was comprised of two multichannel differential mobility analyser (DMA) columns, one for each polarity. Both columns had a software–controlled sample preconditioning unit which allowed the instrument to switch between detecting naturally charged species or uncharged particles. The sheath air flow rate was approximately 60 L $min^{-1}$ and the total sample flow rate was 54 L $min^{-1}$ (divided equally between both DMAs). The time resolution per com-

plete distribution was five–minutes.

#### 2.3.2. Custom–built mobility particle size spectrometer

The principle of the mobility particle size spectrometer (MPSS) is described in detail by Wiedensohler et al. (2012). A custom–built MPSS was used to measure the PNSD (from 5 to 800 nm) for the duration of the measurement campaign. The instrument was comprised of a

bipolar diffusion charger ($^{85}$Kr neutraliser), a Vienna–type DMA (electrode length 280 mm), and a condensation particle counter (CPC model 3772, TSI Inc., USA). The sheath air flow rate (5 L $min^{-1}$) to sample air flow rate (1 L $min^{-1}$) was operated at a ratio of 5:1. Both the aerosol sample flow and sheath air flow were actively dried. Particle losses were quantified and accounted for in the final size distribution. The time resolution for one combined upscan

and downscan was ten–minutes, and the instrument alternated between measuring the total PNSD and the non-volatile PNSD, giving a measurement of the total PNSD every twenty minutes.

#### 2.3.3. Other instrumentation

A nitrate Chemical Ionisation–Atmospheric Pressure interface–Time of Flight mass spectrom-

eter (nitrate CI–APi–ToF) was used to measure neutral $H_2SO_4$ and $(H_2SO_4)nHSO_4$- clusters for the duration of the measurement campaign. The instrument is highly sensitive to strongly acidic compounds, as well as compounds with two hydrogen bond donor groups in the gas phase



(Hyttinen et al., 2015). The front end consists of a chemical ionisation system where a ca. 8 L min$^{-1}$ sample flow is drawn in through a 1 m length ¾" OD stainless steel tube, where it enters

an ionising chamber. Inside the chamber, a secondary flow is run parallel and concentric to the sample flow, rendering the reaction chamber effectively wall–less. A 10 cm$^3$ min$^{-1}$ flow of a carrier gas (in this case, N$_2$) is passed over a reservoir of liquid HNO$_3$, entraining vapour which is subsequently ionised to NO$_3^-$ via an X–ray source. The nitrate ions are then guided into the sample flow by an electric field where they charge molecules by clustering or proton transfer.

The sample enters the critical orifice at the front end of the instrument at 0.8 L min$^{-1}$ and are guided through a series of differentially pumped chambers before they reach the ToF analyser. Data analysis was carried out in the Igor Pro 9. Dried and filtered compressed air was used for the sheath flows.

The instrument was calibrated with respect to sulphuric acid (Kürten et al., 2012). The quantification of sulphuric acid in the nitrate CI–APi–ToF is as follows:

$$[H_2SO_4] = C \times \ln\left(1 + \left(\frac{H_2SO_4NO_3^- + HSO_4^-}{\sum_{n=0-2}(HNO_3)_nNO_3^-}\right)\right) \quad (1)$$

where C is a calibration constant, here $1.07 \cdot 10^9$ cm$^{-3}$ for the instrument. Presuming that all collisions between analyte *A* and the reagent ion result in charging via clustering or deprotona-

tion, the production of charged analytes will continue at the kinetic limit for H$_2$SO$_4$. Blanks were performed mid-campaign. Blank signals were negligible for all compounds of interest.

**2.4. Condensation sink, formation and growth rates**

NPF events were identified visually based on the time evolution of the PNSD plotted as contour plots using the criteria of Dal Maso et al. (2005). Each plot contained data spanning 24 hours

and ranging from 0.8–42 nm (charged species from the NAIS) and 3–800 nm (neutral particles from the NAIS and custom–built MPSS, combined). Each was plotted using a perceptually uniform, high contrast colour palette (Mikhailov, 2019).

The condensation sink (CS) represents the rate at which a vapour phase molecule will collide

with pre–existing particle surface, and was calculated from the size distribution data as follows (Kulmala et al., 2012):

$$CS = 2\pi D \cdot \sum_{d_p} \beta_{m,d_p} \cdot d_p \cdot N_{d_p} \quad (2)$$

where D is the diffusion coefficient of the diffusing vapour (assumed sulphuric acid), $\beta_m$ is a transition regime correction, $d_p$ is particle diameter, and $N_{d_p}$ is the number of particles at





diameter $d_p$. The formation rate of new particles at size dp ($J_{dp}$) is calculated as follows, presuming a homogeneous airmass:

$$J_{d_p} = \frac{dN_{d_p}}{dt} + CoagS_{d_p} \cdot N_{d_p} + \frac{GR}{\Delta d_p} \cdot N_{d_p} \tag{3}$$

where the first term on the right–hand side represents the rate at which particles enter the size $d_p$, and the second term refers to losses from this size by coagulation ($CoagS_{dp}$ being the coag-

ulation sink at size $d_p$, and $N_{dp}$ being the number of particles at size $d_p$, calculated according to Cai and Jiang (2017), with the third term referring to losses from this size by growth. When calculating the formation rate, instead of using a single particle size, a range is used. In this paper we use two ranges, 3–7.5 nm, and 7.5–22 nm for consistency with the size–cuts used for the rest of the analyses. The formation rate of charged particles involves two additional terms,

and is as follows:

$$J_{d_p}{}^{\mp} = \frac{dN_{d_p}{}^{\mp}}{dt} + CoagS_{d_p} \cdot N_{d_p}{}^{\mp} + \frac{GR}{\Delta d_p} \cdot N_{d_p}{}^{\mp} + \alpha \cdot N_{d_p}{}^{\mp} \cdot N_{<d_{p-upper}}{}^{\pm} - \beta \cdot N_{d_p} \cdot N_{<d_{p-lower}}{}^{\mp} \tag{4}$$

Where the fourth term accounts for the loss of charged particles due to their recombination with other charged species of the opposite polarity below the upper bound of *dp*, and the fifth term accounts for the gain of charged particles caused by the attachment of charged species

below the lower bound of *dp* with neutral clusters (Yan et al., 2018). The growth rate (GR) of new particles, which is the change of *dp* over time, is here calculated by the mode–fitting method (Kulmala et al., 2012).

## 3.   RESULTS AND DISCUSSION

### 3.1. Number concentrations of charged particles

Error! Reference source not found. shows a statistical summary of small, intermediate, and l arge charged particle concentrations at Leipzig–TROPOS. Mean number concentrations of small charged particles (0.8–1.6 nm) were 302 and 161 cm$^{-3}$ for positive and negative polarities, respectively. Observed concentrations are comparable with, albeit on the lower end of, the typical tropospheric range reported by Hirsikko et al. (2011). The comparatively low concentration

are in line with the higher coagulation sink for small particles in the urban environment, which is expected to reduce the average concentration. What is immediately apparent is the stark difference in number concentrations between positively and negatively charged species. This dissimilarity is notably pronounced in their respective size distributions (see Error! Reference s ource not found.). Similar disparities between small charged particles of opposing polarities

have been documented in the literature. A measurement campaign in Saare County, Estonia



between July and September 1984 reported mean concentrations of positively and negatively charged small species of 261 and 173 cm⁻³, respectively (Hõrrak, 1987). The imbalance is believed to be caused by the Earth's negatively charged surface impacting the distribution of charged species, referred to as the electrode effect (Hoppel, 1967).

**Table 1: Statistical summary of small (0.8–1.6 nm), intermediate (1.6–7.5 nm), large (7.5–22 nm), and total charged particle number concentrations (0.8–42 nm) per cm⁻³. Data coverage: 27th July 2022 14:00 to 25ᵗʰ August 2022 08:00 (UTC) using hourly means.**

|  | Mean | Median | 5–95% |
|---|---|---|---|
| **Small charged particles (+)** | 301.9 | 294.4 | 193.3–447.3 |
| **Small charged particles (-)** | 160.6 | 155.4 | 90.9–244.8 |
| **Intermediate charged particles (+)** | 44.8 | 26.2 | 12.8–143.2 |
| **Intermediate charged particles (-)** | 42.9 | 22.4 | 7.7–162.0 |
| **Large charged particles (+)** | 207.6 | 149.6 | 44.4–625.2 |
| **Large charged particles (-)** | 212.7 | 154.8 | 55.2–617.2 |
| **Total charged particles (+)** | 832.9 | 767.6 | 431.6–1,531.7 |
| **Total charged particles (-)** | 702.0 | 619.2 | 335.8–1,395.8 |

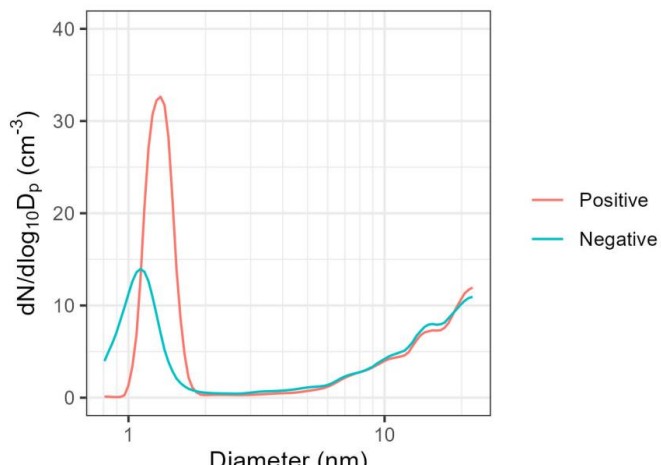

**Figure 2: Size distribution of positive and negatively charged particles between 0.8 and 22 nm. Data coverage: 27ᵗʰ July 2022 14:00 to 25 August 2022 08:00 (UTC).**

Mean concentrations of intermediate charged particles (1.6–7.5 nm), on the other hand, were comparatively very low. However, they were present in substantially larger concentrations on

NPF event days compared to non–NPF event days (see **section 3.5.**). Mean number





concentrations of intermediate charged species were 45 and 43 cm$^{-3}$ for positive and negative polarities, respectively. Observations are very similar to annual mean concentrations (35–40 cm$^{-3}$ for each polarity) recorded between April 2010 and November 2011 in Tartu, Estonia by Tammet et al. (2014). Though, they are approximately 1.5–2.1 times higher (depending on

polarity) than mean concentrations recorded between June 2009 and October 2010 in Paris, France by Dos Santos et al. (2015). Observed variability may be explained by proximity to and density of the surrounding transport infrastructure (see **section 3.2.**), photochemical processes (see **section 3.5.**), and length of campaign period.

Much like intermediate charged particles, there were no significant differences in mean concentrations between the opposing polarities of large charged species (7.5–22 nm). They were also present in much larger concentrations on NPF event days compared to non–NPF event days (see **section 3.5.**). However, mean concentrations of large charged particles (during the whole campaign) were considerably higher than intermediate charged species. Mean number

concentrations were 208 and 213 cm$^{-3}$ for positive and negative polarities, respectively, and were approximately 4.6–5.0 times higher (depending on polarity) than intermediate charged particles. Observed differences between these charged particle mobility classifications may be attributed to the respective impact of local air pollution (see **section 3.2.**).

**3.2. Diurnal cycles of charged particles**

**Figure 3** shows the mean diurnal cycles of small, intermediate, and large charged particles at Leipzig–TROPOS. Small charged particle concentrations peaked in the early morning (03:00–04:00 (UTC)), decreased into the afternoon (11:00–13:00), and increased into the night. Such observations are comparable to other studies in Pune, India (Dhanorkar and Kamra, 1994),

Tumbarumba, Australia (Suni et al., 2008), and Paris, France (Dos Santos et al., 2015) and may be attributed to fluctuations in boundary layer mixing height and the accumulation of radioactive gases (e.g. radon decay). Concentrations of small charged species increased prior to the below–mentioned peaks in intermediate and large charged particle concentrations and decreased thereafter. Diurnal cycles suggest that small charged species arise primarily from nat-

ural processes and are quickly lost via recombination and attachment to larger aerosols. As cosmic ray production is constant at the earths surface, this natural production is in-line with production due to radioactive decay.



Intermediate charged particle concentrations peaked several hours after the initial peak in small

charged species (08:00) and again later in the day (20:00). Similarly, large charged particle concentrations peaked some hours later (10:00) and lesser peaks were observed in the morning (05:00–06:00) and in the evening/night–time (21:00). 'Midday' peaks in both classifications coincided with intense solar radiation (see **section 3.4.**) and occurred when NPF events were observed (see **section 3.5.**). Lesser peaks coincided with busy road traffic periods and eco-

nomic activities, known to emit high quantities of positive and negative charged species (Jayaratne et al., 2010, 2014). Pollution–related peaks appeared more pronounced in the large charged fraction. Diurnal cycles suggest photochemistry and local air pollution dominate intermediate and large charged particle production, with the latter contributing more significantly to large charged particle concentrations.




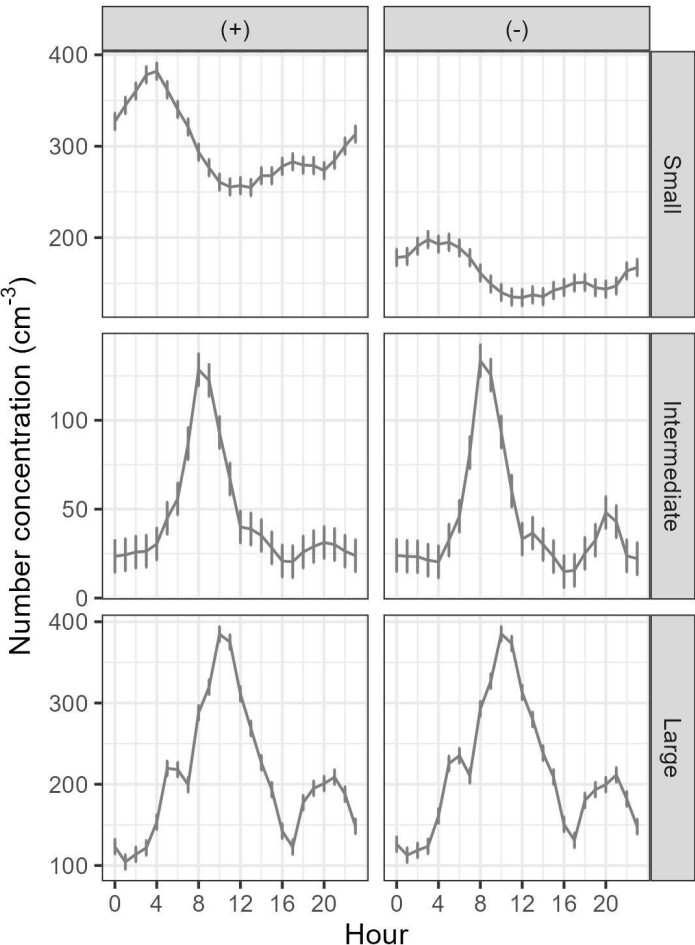

**Figure 3: Diurnal cycles of small (0.8–1.6 nm), intermediate (1.6–7.5 nm), and large (7.5–22 nm) charged particles. The vertical lines represent the standard error of the mean. Data coverage: 27th July 2022 14:00 to 25 August 2022 08:00 (UTC) using hourly means.**

### 3.3. Frequency of new particle formation

Measurement days were classified into three categories: NPF event, undefined, and non–NPF event according to methods described by Dal Maso et al. (2005). NPF event days were classified as such when days showed both particle formation and growth. Equally, undefined days were assigned when days satisfied some but not all of the aforementioned criteria (i.e. a new but non–persistent mode or no clear signs of growth). Lastly, non–NPF event days were grouped as such when the charged and neutral PNSD data showed no clear indication of new particle formation. A total of 9 NPF event, 6 undefined, and 15 non–NPF event days were identified across the 30–day measurement campaign at Leipzig–TROPOS. The frequency of



NPF event days (30%) was comparable with frequencies from long–term analysis (Bousiotis
et al., 2021).

### 3.4. Meteorology and charged particles

Error! Reference source not found. shows the correlation coefficients between meteorological v
ariables and charged particle mobility classifications at Leipzig–TROPOS. Solar radiation and
air temperature exhibited negative correlations with small charged particles but positive corre-
lations with intermediate and large charged species. Conversely, relative humidity showed pos-
itive correlations with small charged species and negative correlations with intermediate and
large charged particles. These trends align with expectations, considering the well–established
positive relationship between solar radiation and air temperature, coupled with their inverse
relationships with relative humidity. Fluctuations in boundary layer mixing height, and the ac-
cumulation of radioactive gases, discussed in **section 3.2.**, are believed to have influenced the
small charged fraction. Mixing height is a dynamic parameter impacted by a variety of factors,
including time of day and weather conditions. The parameter is habitually related to air tem-
perature, with cooler morning temperatures theoretically limiting vertical mixing and inadvert-
ently enhancing small charged particle concentrations. Studies suggest that solar radiation, par-
ticularly in the ultraviolet spectrum, can play a significant role in air ionisation (Jiang et al.,
2018). Observed correlations imply that solar radiation may have contributed to intermediate
and large charged particle concentrations through photoionisation.



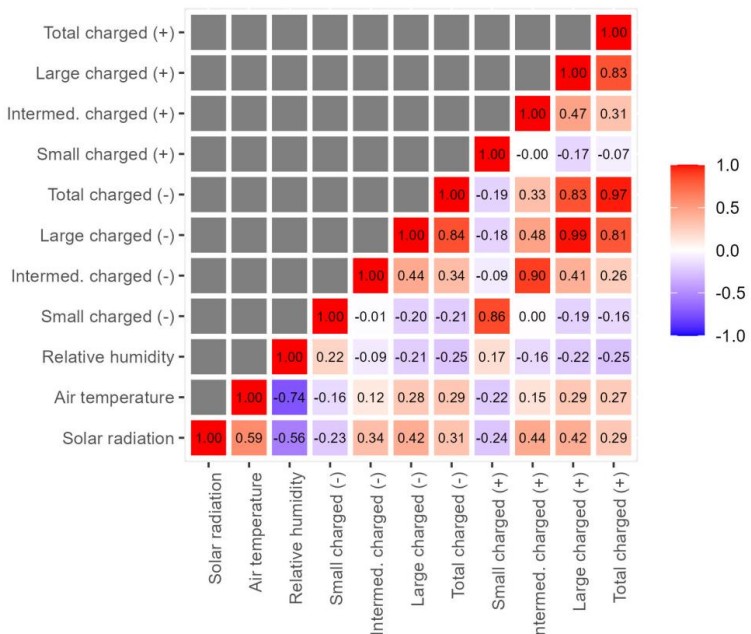

**Figure 4: Pearson correlation matrix heatmap of meteorological variables (solar radiation, air temperature, and relative humidity) and small, intermediate, large, and total charged particles (of both polarities). Warm colours (red) represent positive correlations, and cold colours (blue) represent negative correlations. Correlation strength ranges from -1 to +1. The shade indicates the strength of the correlation, with darker shades indicating stronger correlations. Data coverage: 27th July 2022 14:00 to 25 August 2022 08:00 (UTC) using hourly means.**

**Figure 5** shows the mean diurnal cycles of meteorological variables during NPF event, undefined, and non–NPF event days at Leipzig–TROPOS. Throughout the day, solar radiation and air temperature generally exhibited higher values, while relative humidity was lower on NPF event days compared to non–NPF event days. Notably, during the early morning hours, air temperatures were slightly lower on NPF event days, with more pronounced differences observed compared to undefined days. In the literature, the role of air temperature in NPF in the urban atmosphere is ambiguous, however, the intensity of solar radiation is thought to play an important role in whether atmospheric NPF takes place or not (Kerminen et al., 2018). Higher temperatures can increase the evaporation rate of molecular clusters, potentially decreasing particle formation rates (Lee et al., 2019). However, charged particles may play a significant role in stabilising clusters, particularly at slightly warmer temperatures (Lee et al., 2019).



Relative humidity, on the other hand, tends to be lower on NPF event days compared to non–NPF event days – with several possible reasons for this apparent close connection in the liter-

ature (Kerminen et al., 2018). Diurnal cycles reveal that NPF events coincided with intense solar radiation, substantial air temperature variations between morning and afternoon hours, and a significant decrease in relative humidity from morning to afternoon.

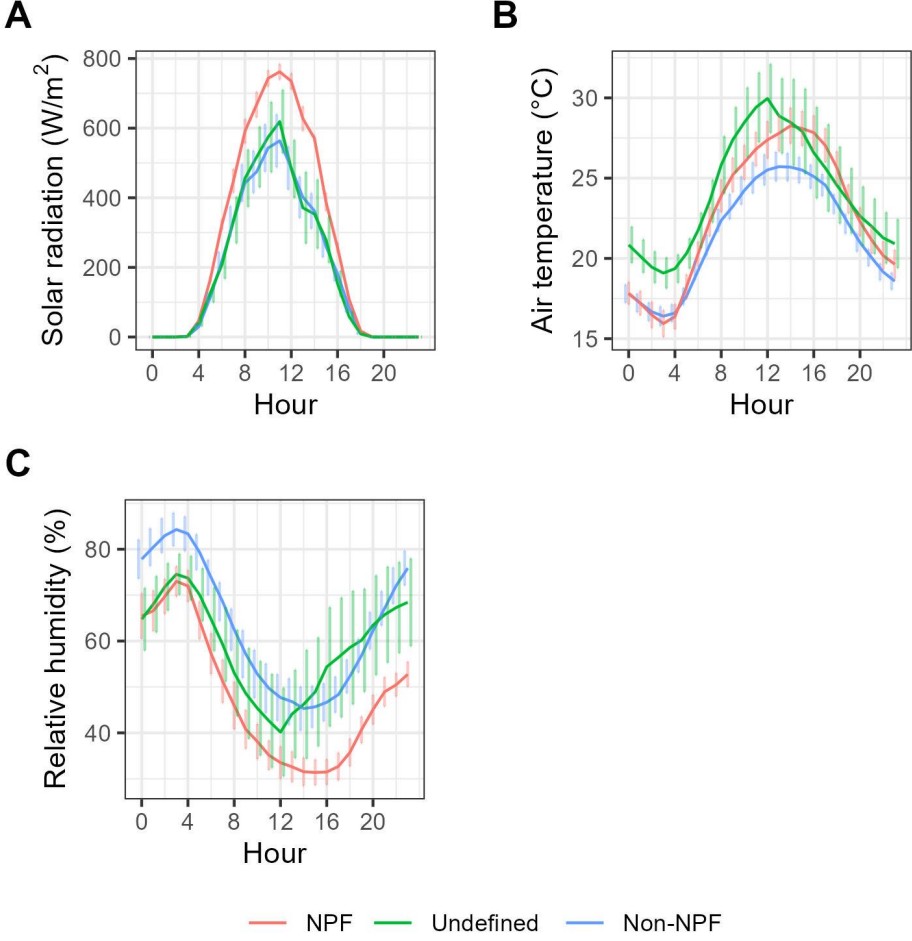

**Figure 5: Diurnal cycles of (A) solar radiation, (B) air temperature, and (C) relative humidity on new particle formation (NPF) event, undefined, and non–NPF event days. The vertical lines represent the standard error of the mean. Data coverage: 27$^{th}$ July 2022 14:00 to 25 August 2022 08:00 (UTC) using hourly means.**



**3.5. Diurnal cycles of charged particles during new particle formation**

Error! Reference source not found. shows the mean diurnal cycles of small, intermediate, and l arge charged particles on NPF event, undefined, and non–NPF event days at Leipzig–TROPOS. On NPF event days, diurnal maxima of small charged particles were observed between 03:00 and 06:00 (UTC) and minima between 12:00 and 14:00. Diurnal maxima of intermediate and large charged species were observed at 08:00 and 10:00, respectively. Time–gaps

between maximum concentrations of intermediate and large charged particles (approximately two hours) likely indicate growth between size classifications, a phenomenon not observed to the same degree on non–NPF event days (alternative graphic presented in **Figure S1**). Comparable time–gaps have been observed in both urban (Dos Santos et al., 2015) and rural (Hõrrak et al., 2003) settings. Small charged particle concentrations were generally lower on NPF event

days compared to non–NPF event days, consistent with findings in rural areas (Gagné et al., 2010; Hõrrak et al., 2003). In contrast, maximum concentrations of intermediate and large charged species were approximately 4.0–4.4 and 3.6–3.7 times higher (depending on polarity), respectively, on NPF event days compared to non–NPF event days. Diurnal cycles suggest a substantial increase in intermediate and large charged particles associated with NPF event days

at Leipzig–TROPOS.



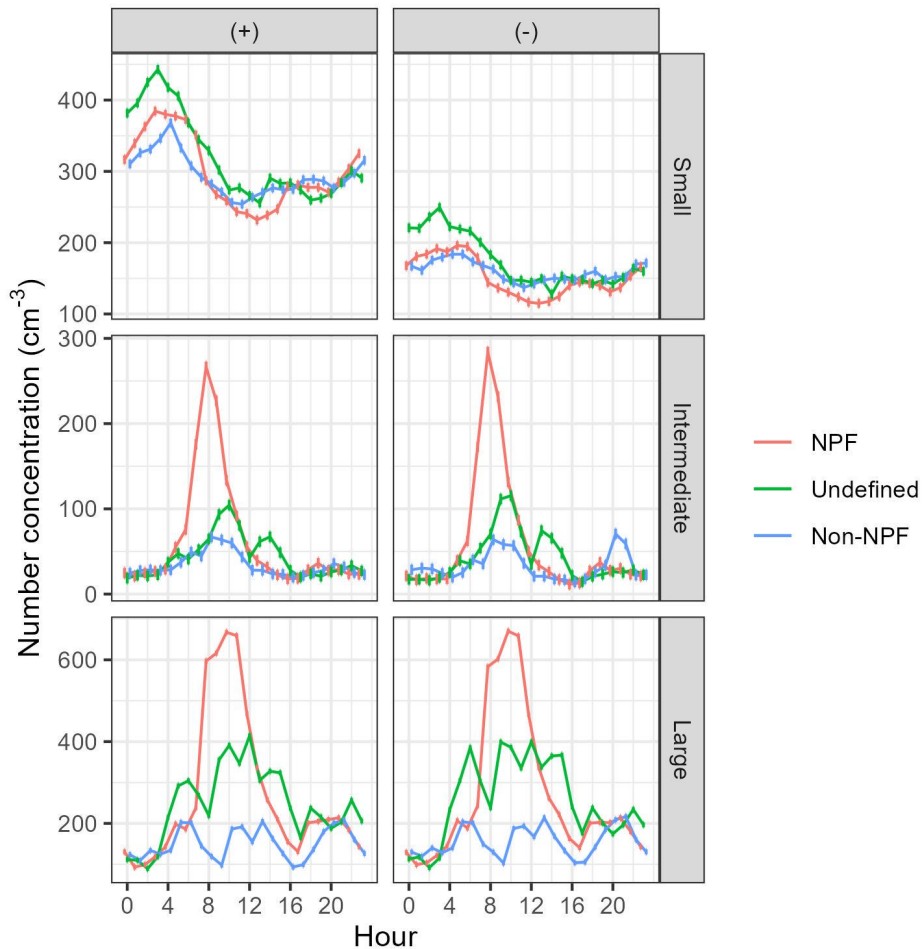

**Figure 6: Diurnal cycles of small (0.8–1.6 nm), intermediate (1.6–7.5 nm), and large (7.5–22 nm) charged particles on new particle formation (NPF) event, undefined, and non–NPF event days. The vertical lines represent the standard error of the mean. Data coverage: 27th July 2022 14:00 to 25 August 2022 08:00 (UTC) using hourly means.**

**Figure 7** shows the mean diurnal cycles of black carbon (BC), sulphuric acid ($H_2SO_4$) dimer, and condensation sink (CS) concentrations on NPF event, undefined, and non–NPF event days at Leipzig–TROPOS. BC concentrations were generally lower in the morning and into the early evening, and noticeably higher in the late evening/night–time, on NPF event days compared to non–NPF event days. Morning and late evening/night–time peaks occurred synchronously with peaks in large charged particles. BC is often used as a proxy for traffic–related air pollution and other combustion–related activities. Peaks in BC were also observed in the CS due to the



high surface area of BC–containing particles. Maximum $H_2SO_4$ dimer concentrations peaked synchronously with intermediate charged particle concentrations. In the nitrate CI–APi–ToF, the $H_2SO_4$ dimer is a representation of atmospheric $H_2SO_4HSO_4^-$, larger atmospheric sulphuric acid–base clusters which undergo evaporation due to chemical ionisation, and some ion-molecule pairing in the front of the CIMS inlet (Almeida et al., 2013) and is considered a good proxy

for the occurrence of NPF in urban environments (Yao et al., 2018). $H_2SO_4$ dimer is highest on NPF days, while BC and CS are low. A CS peak approximately five hours after the $H_2SO_4$ dimer peak on NPF event days reflects the growing mode of new particles contributing appreciably to surface area. Undefined and non-NPF events are observed when $H_2SO_4$ dimer is low. Undefined events are seen when BC and CS is high, likely due to traffic emissions, and non-

events are observed when BC and CS are lower. Non-NPF days are possibly observed on these days due to low source strengths of precursors. Observed similarities between diurnal cycles of charged particles and those of BC and $H_2SO_4$ dimer provide evidence of multiple sources for charged species in our data.





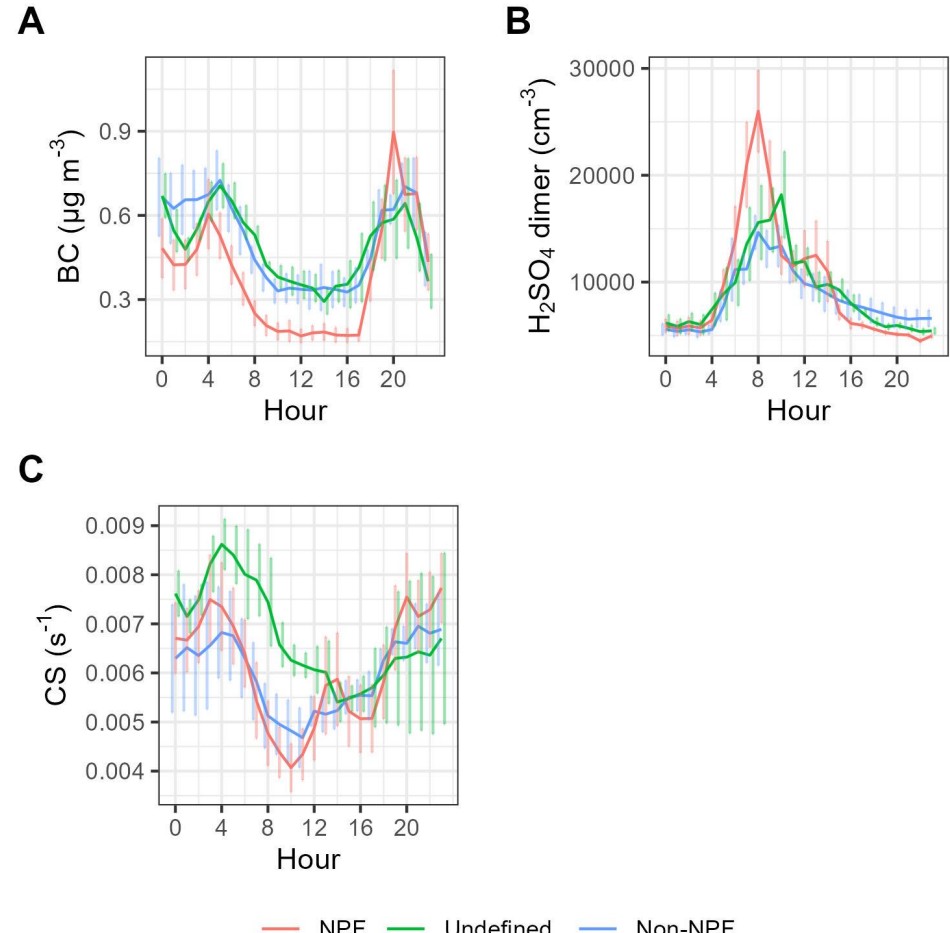


**Figure 7: Diurnal cycles of (A) black carbon (BC), (B) sulphuric acid (H₂SO₄) dimer, and (C) condensation sink (CS) on new particle formation (NPF) event, undefined, and non–NPF event days. The vertical lines represent the standard error of the mean. Data coverage: 1st August 2022 00:00 to 24th August 2022 11:00 (UTC) using hourly means.**

**3.6. Charged particles and particle formation rates**

**Figure 8** shows the apparent formation rates ($J$s) of 3 and 7.5 nm charged particles (positive and negative polarities, combined; $J_{3-7.5}^{charged}$ and $J_{7.5-22}^{charged}$) and neutral particles (charged and neutral particles, combined; $J_{3-7.5}^{neutral}$ and $J_{7.5-22}^{neutral}$) during NPF event days at Leipzig–TROPOS. Notably, the $J$s of charged and neutral particles generally increased with aerosol

size. The mean $J$s of 3 and 7.5 nm charged particles were 0.020 and 0.093 cm⁻³ s⁻¹, respectively, with mean values of $J_{7.5-22}^{charged}$ approximately 4.7 times higher than $J_{3-7.5}^{charged}$. These compare with mean $J$s of 3 and 7.5 nm neutral particles of 0.622 and 0.673 cm⁻³ s⁻¹, respectively, with





mean values of $J_{7.5-22}^{\text{neutral}}$ approximately 1.1 times than $J_{3-7.5}^{\text{neutral}}$. The aforementioned $J$s are within the observed tropospheric ranges for charged and neutral particles reported by Hirsikko

et al. (2011). When considering the calculated ratios of $J^{\text{charged}} / J^{\text{neutral}}$ in the respective size ranges, the apparent mean contributions of charged particles to 3 and 7.5 nm total particle formation were 5.7 and 12.7%, respectively. The dynamic interplay between charged and neutral particles results in a shifting ratio. Larger charged particles, with their increased size, exhibit a higher likelihood of acting as nucleation sites or aggregating with other charged species includ-

ing ionised gas molecules. Their larger size equates to a greater surface area and a more stable charge. In comparison, smaller charged particles, while possessing higher mobilities due to their reduced size, face a greater susceptibility to ion–ion recombination. This recombination process, wherein two oppositely charged particles combine and neutralise each other, accounted for in equation (4), can impact the abundance of smaller charged species, influencing

their ability to contribute to nucleation and particle formation in the atmosphere. It would be reasonable to view $J_{3-7.5}^{\text{charged}}$ as an upper limit to ion–induced nucleation, while larger charged particles appear to have a substantial contribution from charges acquired subsequently. The apparent contributions are comparable with ranges from other European field sites (1–30%) covering a wide variety of environments reported by Manninen et al. (2010). Nevertheless,

observed ratios of charged to uncharged particles in the size range impacted by NPF suggest charged species play a minor role compared to neutral species in NPF at Leipzig–TROPOS.

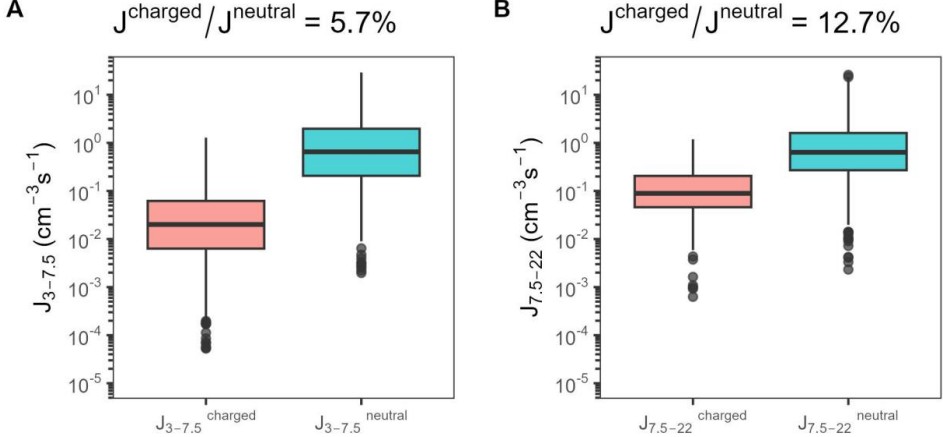

**Figure 8: Apparent formation rates of (A) 3–7.5 nm charged particles (left) and neutral**
**particles (right) and (B) 7.5–22 nm charged particles (left) and neutral particles (right).**
**Calculated from 9 new particle formation (NPF) event days using 10–minute means. The**



**3.7. Charged particle growth rates**

**Figure 9** shows growth rates (*GR*s) of charged particles within diameters 3–7.5 and 7.5–22 nm during NPF event days at Leipzig–TROPOS. Consistent with previous studies (Manninen et al., 2010; Dos Santos et al., 2015; Svensmark et al., 2017), the GR of charged species generally increased with size. This observation is attributed to the Kelvin effect, where the condensa-

tional growth of smaller particles is driven by a limited number of very low volatility compounds. In contrast, the growth of larger particles is influenced by a greater number of molecules, including oxygenated organic molecules (OOMs). Contrary to *J*s, discussed in **section 3.6**, *GR*s of charged species are expected to align more closely with that of neutral particles. Small discrepancies may arise due to the enhanced condensation of ionised gases and improved

coagulation resulting from charge–charge effects (Svensmark et al., 2017). Mean *GR*s of 3–7.5 and 7.5–22 nm charged particles were 4.0 and 5.2 nm h$^{-1}$, respectively. In comparison, Manninen et al. (2010) reported median *GR*s from various European field sites as 4.3 and 5.4 nm h$^{-1}$ for 3–7 nm and 7–20 nm charged species, respectively.

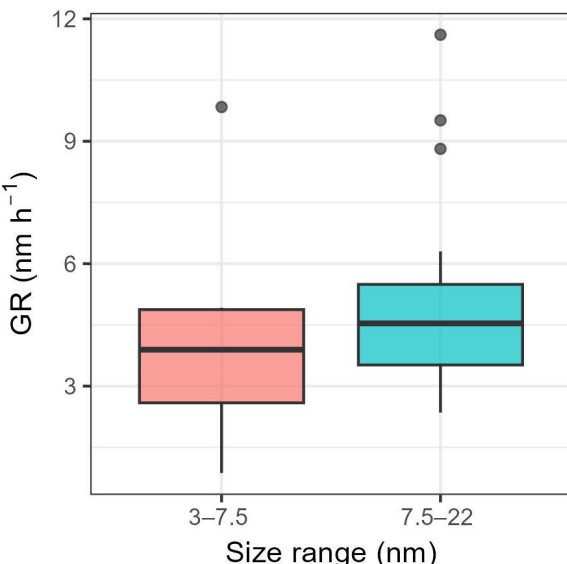


**Figure 9: Growth rates (GR) of 3–7.5 and 7.5–22 nm charged particles. The coloured box represents the middle 50% of the data, with the central horizontal line indicating the**



**median value. The whiskers (vertical lines) extending from the box show the spread of the data. Data points beyond the whiskers show outliers.**

**4. CONCLUSION**

In this study, a detailed analysis of charged particle dynamics at Leipzig–TROPOS was undertaken, aiming to better understand the behaviour of charged species and their contribution to NPF in an urban context. Throughout the measurement campaign, small, intermediate and large charged particles were ever–present. Small charged particle concentrations were consistent

with observations in the existing literature. A clear disparity was evident between positive and negative polarities, attributed to the Earth's electrode effect. Despite these differences, their diurnal cycles were very similar. Small charged particle concentrations peaked in the early morning, decreased into the afternoon, and rose again into the night. These fluctuations are believed to be related to changes in the boundary layer mixing height and the accumulation of

radioactive gases.

Intermediate charged particle concentrations were comparatively low, while large charged particles presented concentrations similar to the small fraction. Variable concentrations were observed in previously published data, possibly linked to photochemical processes, the proximity to and density of the surrounding road transport infrastructure, and length of study period.

Maximum concentrations of intermediate and large charged species were observable in the morning hours, with the latter peaking closer to midday. Local air pollution had a more substantial impact on larger charged particles compared to small and intermediate charged species, indicated by synchronous peaks in black carbon concentrations.


NPF events were identified on 30% of measurement days, occurring under intense solar radiation, significant diurnal temperature fluctuations, and decreasing relative humidity from morning to afternoon. Notably, small charged particle concentrations were typically lower on NPF event days compared to non–NPF event days. Peak concentrations of intermediate and large

charged species were approximately 4.0–4.4 and 3.6–3.7 times higher (depending on polarity), respectively, on NPF event days compared to non–NPF event days. $H_2SO_4$ dimer concentrations were elevated on NPF event days and peaked synchronously with intermediate charged particle concentrations.



The apparent contributions of charged species to 3 and 7.5 nm particle formation were 5.7 and 12.7%, respectively, with mean growth rates of 4.0 and 5.2 nm h$^{-1}$. Both the apparent formation and growth rates of charged particles increased with aerosol size and were found to be comparable with ranges reported in previous studies. The ratio of uncharged to charged nanoparticles and small magnitude of $J_{3-7.5}^{charged}$ suggest that ion–induced processes play a minor role com-
pared to neutral species in NPF at Leipzig–TROPOS.

## DATA AND MATERIALS AVAILABILITY

Data supporting this publication are openly available from the UBIRA eData repository at

https://doi.org/10.25500/edata.bham.00001073.

## AUTHOR CONTRIBUTIONS

Conceptualisation – AR and JB; data curation – AR and JB; formal analysis – AR and JB; funding acquisition – RH; investigation – AR, JB, AK, SB, SI, AK; methodology – AR and JB; project administration – RH; resources – MR, MDM, PM, KW, MM, TT; software – AR and JB; supervision – RH; visualisation – AR and JB; writing (original draft preparation) –
AR; writing (review & editing) – AR, JB, DB, ZS, AK, SB, MR, MDM, PM, KW, and RH.

## COMPETING INTERESTS

The authors declare that they have no conflict of interest.

## ACKNOWLEDGEMENTS

University of Birmingham would like to express sincere appreciation to TROPOS for their
gracious hosting and invaluable assistance during our measurement campaign.

## FUNDING

This study was supported by funding from UK Natural Environment Research Council (NERC NE/V001523/1).



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





**TABLE LEGEND**

**Table 2:** Statistical summary of small (0.8–1.6 nm), intermediate (1.6–7.5 nm), large (7.5–22 nm), and total charged particle number concentrations (0.8–42 nm) per cm$^{-3}$. Data coverage: 27th July 2022 14:00 to 25$^{th}$ August 2022 08:00 (UTC) using hourly means.


**FIGURE LEGENDS**

**Figure 10:** Location of the TROPOS site (red marker), approximately 4 km northeast of Leipzig city centre.


**Figure 11:** Size distribution of positive and negatively charged particles between 0.8 and 22 nm. Data coverage: 27$^{th}$ July 2022 14:00 to 25 August 2022 08:00 (UTC).

**Figure 12:** Diurnal cycles of small (0.8–1.6 nm), intermediate (1.6–7.5 nm), and large (7.5–22 nm) charged particles. The vertical lines represent the standard error of the mean. Data coverage: 27$^{th}$ July 2022 14:00 to 25 August 2022 08:00 (UTC) using hourly means.

**Figure 13:** Pearson correlation matrix heatmap of meteorological variables (solar radiation, 740 air temperature, and relative humidity) and small, intermediate, large, and total charged particles (of both polarities). Warm colours (red) represent positive correlations, and cold colours (blue) represent negative correlations. Correlation strength ranges from -1 to +1. The shade indicates the strength of the correlation, with darker shades indicating stronger correlations. Data coverage: 27$^{th}$ July 2022 14:00 to 25 August 2022 08:00 (UTC) using hourly means.


**Figure 14:** Diurnal cycles of (A) solar radiation, (B) air temperature, and (C) relative humidity on new particle formation (NPF) event, undefined, and non–NPF event days. The vertical lines represent the standard error of the mean. Data coverage: 27$^{th}$ July 2022 14:00 to 25 August 2022 08:00 (UTC) using hourly means.


**Figure 15:** Diurnal cycles of small (0.8–1.6 nm), intermediate (1.6–7.5 nm), and large (7.5–22 nm) charged particles on new particle formation (NPF) event, undefined, and non–NPF event days. The vertical lines represent the standard error of the mean. Data coverage: 27$^{th}$ July 2022 14:00 to 25 August 2022 08:00 (UTC) using hourly means.


**Figure 16:** Diurnal cycles of (A) black carbon (BC), (B) sulphuric acid (H$_2$SO$_4$) dimer, and (C) condensation sink (CS) on new particle formation (NPF) event, undefined, and non–NPF event days. The vertical lines represent the standard error of the mean. Data coverage: 1$^{st}$ August 2022 00:00 to 24$^{th}$ August 2022 11:00 (UTC) using hourly means.


**Figure 17:** Apparent formation rates of (A) 3–7.5 nm charged particles (left) and neutral particles (right) and (B) 7.5–22 nm charged particles (left) and neutral particles (right). Calculated from 9 new particle formation (NPF) event days using 10–minute means. The coloured rectangle represents the middle 50% of the data, with the central horizontal line indicating the 765 median value. The whiskers (vertical lines) extending from the rectangle show the spread of the data. Data points beyond the whiskers show outliers.

**Figure 18:** Growth rates (GR) of 3–7.5 and 7.5–22 nm charged particles. The coloured box represents the middle 50% of the data, with the central horizontal line indicating the median



value. The whiskers (vertical lines) extending from the box show the spread of the data. Data
       points beyond the whiskers show outliers.