# Peer review of "The behaviour of charged particles (ions) during new particle"

_EGUsphere, 2024_

## Author Comment (AC1)

**The behaviour of charged particles (ions) during new particle formation events in urban Leipzig (Germany). Response to reviewers.**

Note: Review comments are displayed in plain text, responses to those comments are displayed in blue and sections that have been added to the text are coloured *green (and italicised)* We thank the reviewers for their insightful comments and provide responses below.

**Reviewer: 1**

In this manuscript, the authors investigated the number size distribution of atmospheric ions observed in Leipzig, Germany. The authors showed that ions classified into different size ranges have different diel behaviors and explained such behaviors in association with other atmospheric parameters.

In general, I see nothing wrong in this manuscript, where most of the explanations are scientifically sound. However, I am a little concerned about the significance of the findings presented in this manuscript. I didn't see enough new insights, except for the new locations. Since the authors deployed a nitrate CIMS during the measurement, if they can associate the concentration and composition changes of gaseous species with the variability of ions, it is possible to bring this study to a new level.

Thank you for the comments. We have now used the CIMS data more extensively to inform our data analysis. We conclude that there are two primary sources of ions >3 nm in our data: these are primary emissions and NPF. We hope the below additional data and discussion emphasises this point and shows that we satisfy the novelty criteria for the journal.

In the abstract:

[revised manuscript text omitted]

*."*

Technical issues:

1. there are several places (mostly at the beginning of paragraphs) show format issues.

Thanks for pointing this out. These have been amended.

2. I think the nighttime high concentration of small ions is due to its connection with boundary
layer dynamics, as well as the competition between particles of different sizes in taking up the
ions. Without solid proof, it is not convincing to say the diel pattern of smallest ions are due
to radioactive decay. (Line 255).

Very true. We have amended this section as follows to include more discussion of radon.

*"Cosmic ray intensity is fairly constant throughout the lower atmosphere (Mercer and Wilson, 1965),*
*while the variations in radon concentrations is attributable to boundary layer dynamics (Čeliković et*
*al., 2023). The diurnal variation which we observe is therefore likely to be a combination of boundary*
*layer height changes affecting the radon concentrations, and variations in particle number surface*
*area altering coagulation rates due to both boundary layer height changes and primary and*
*secondary particle emissions."*

**134 Reviewer: 2**

Rowell et al. have studied the concentrations, growth rates and formation rates of small ions,
intermediate ions, and large ions in Leipzig, Germany, during a month long campaign in summer
2022. They paid special attention to the charged particles and air ions during NPF event days.

One issue becomes apparent immediately. The size ranges, which are used in this study to classify
ions into small ions, intermediate ions and large ions are, to my understanding, based on mass
diameters. However, the diameters used in this study are mobility diameters. It is very possible that
the impacts of this on the results themselves are minor. However, the concentrations of sub-2 nm ions
can be considerably higher than those of above 2 nm ions, which might impact the intermediate ion
concentrations used in this study.

Some factors, such as missing information from the methods section and multiple errors with the
references to tables/figures, give an impression of a rushed work. Quite a few relatively minor issues
could also be identified. In addition, I found it a bit difficult to understand some of what had been
done, i.e., how the formation rates and growth rates had actually been determined for the ions.

The writing itself mostly does it job, although at places I found the analysis and argumentation
difficult to understand and follow. The amount of references to other studies was a bit lacking at some parts. In addition, I find both the results and analysis in general a bit lacking in depth and novelty.
There is potential for more, even with the data the authors likely possess already, some suggestions to
which I give below in the more detailed comments.

Despite these issues, I have no doubt many readers of ACP, myself included, would find these results
(with the issues addressed) interesting. Therefore, after the comments below have been sufficiently
addressed, I would consider the study worth publishing. However, I find that with the current
contents, the study might be more suitable as a measurement report than as a research article.

We thank the reviewer for the extensive comments they've made, and agree that addressing them has
strengthened the manuscript. Specific replies to comments are below.

**Specific comments**

**Abstract**

Line 23: I have concerns regarding this size classification. The limits of the size classification used by
Tammet (2006), were based on mass diameters (see J. Aerosol Sci., 26 (1995), pp. 459-475). Here,
mobility diameters are used instead, while the diameter ranges are the same. While unlikely to have a
major effect on the main results, the diameter limits should be reconsidered.

We agree these are erroneous. For consistency with BSMA measurements, we have converted all
sizecuts to the appropriate sizecuts following the suggestion of Ku & de la Mora (2009), and making
the alteration using the effective gas diameter of 0.3 nm. We have reproduced all figures and
reworked all relevant parts of the text (not included below for length considerations).

Ku, B. K., & de la Mora, J. F. (2009). Relation between Electrical Mobility, Mass, and Size for
Nanodrops 1–6.5 nm in Diameter in Air. *Aerosol Science and Technology*, *43*(3), 241–249.
https://doi.org/10.1080/02786820802590510

In addition, here, and later in the manuscript, 0.8-1.6 nm ions are referred to as small charged
particles. This is not accurate, as some of the ions in this size range could be large, charged molecules.
As such, referring to them as small (air) ions instead of small charged particles would be more
accurate. Alternatively, it should be defined that small charged particles can include also large
charged molecules. Please revise accordingly.

Thanks for this comment. We use charged *particles* specifically because in most fields, ion refers
only to charged atoms or molecules, so using it to refer to a charged aerosol seemed strange. We
opted to use particle as it is a catch-all term that can refer to atoms, particles, or aerosols. We specify
this in the following in the methods:

*"Here, we refer to all charged species measured by the NAIS as "charged particles", which includes*
*charged aerosols, as vwell as charged molecules and charged clusters of molecules."*

2. Line 30: the reason why small ion concentrations are lower and intermediate/large ion
concentrations higher on NPF event days could be mentioned.

We do not know the exact reason for lower small charged particle concentration, however, we state
the following

*"Small charged particles were primarily associated with radioactive decay during the early hours,*
*and are unrelated to primary emissions or NPF"*

3. Line 32: here, and also many times later in the manuscript, the phrase charged (or neutral species)
is used. I am not sure if it is an accurate phrasing to use for charged (neutral) particles, with varying,
non-uniform chemical makeup. I suggest using particles instead to avoid any confusion.

Very true! We have edited all instances of "charged species" to "charged particles".

Alongside these comments, we have reworked other areas of the abstract in line with other reviewer
comments. It now reads as follows:

*"Air ions are electrically charged particles in air. They are ubiquitous in the natural environment and*
*affect the earths radiation budget by accelerating the formation and growth of new aerosol particles.*
*Despite this, few datasets exist exploring these effects in the urban environment. A Neutral cluster and*
*Air Ion Spectrometer was deployed in Leipzig, Germany, to measure the number size distribution of*
*charged particles from 0.8 to 42 nm, between July 27th and August 25th 2022. Following previous*
*analyses, charged particles were classified into small (0.8–1.6 nm), intermediate (1.6–7.5 nm), and*
*large (7.5–22 nm) fractions by mass diameter and their mean concentrations (sum of positive and*
*negative polarities) during the campaign were 405, 71.6, and 415 $cm^{-3}$, respectively. The largest*
*peaks in intermediate and large ions were explained by NPF, with intermediate ions correlating well*
*with sulphuric acid dimer. Smaller morning and evening peaks were coincident with black carbon*
*concentrations, and attributed to primary emissions. NPF events, observed on 30% of days, coincided*
*with intense solar radiation and elevated sulphuric acid dimer. Small charged particles were*
*primarily associated with radioactive decay during the early hours, and are unrelated to primary*
*emissions or NPF. The apparent contributions of charged particles to 3 and 7.5 nm particle formation*
*rates were 5.7 and 12.7%, respectively, respectively, with mean growth rates of 4.0 nm $h^{-1}$ between 3-*
*7.5 nm and 5.2 nm $h^{-1}$ between 7.5-22 nm. The ratio of charged to total particle formation rates at 3*
*nm suggests a minor role for charged particles in NPF. We conclude that NPF is a primary source of*
*>3 nm ions in our data, with primary emissions being the major source in the absence of NPF."*

**Introduction**

4. Line 43-44, line 51-52: a reference is needed.

We include a reference to Seinfeld and Pandis (2016) here, as they extensively discuss the variability
of aerosols.

*"Seinfeld, J.H. and Pandis, S.N. (2016) Atmospheric Chemistry and Physics: From Air Pollution to*
*Climate Change. John Wiley & Sons, Hoboken."*

5. Line 58: I do not understand what "persist as a source of charge" means.

We update this sentence to read as follows

*"Following nucleation and the formation of stable new particles, ion-induced condensation can*
*accelerate particle growth (Svensmark et al., 2017)."*

6. Line 74, line 76: vague phrasing. The wording "In other remote locations" suggest that the
locations studied by Manninen et al. (2010) were all remote and that there was no overlap between the
two studies (accuracy of the latter I cannot confirm, see comment below). "In other urban locations"
is similarly unclear.

We update this sentence to read as follows

*"Manninen et al. (2010) found that contributions of ion–induced nucleation to total particle*
*formation at 2 nm were typically in the range of 1–30% between 12 field sites across Europe. In*
*remote locations, Kulmala et al. (2010) found that contributions were typically significantly less than*
*10% in Hyytiälä (Finland), Hohenpeissenberg (Germany), and Melpitz (Germany). In urban*
*locations, contributions were observed at approximately 1.3% at 1.5/2 nm in Helsinki, Finland*
*(Gagné et al., 2012) and 10% at 3 nm in Brisbane, Australia (Pushpawela et al., 2018)."*

7. Line 74: Kulmala et al. (2010) is not in the reference list, or at least I cannot find it.

Apologies. This is the proper reference, which has been added:

*"Kulmala, M., Riipinen, I., Nieminen, T., Hulkkonen, M., Sogacheva, L., Manninen, H. E., Paasonen, P., Petäjä, T., Dal Maso, M., Aalto, P. P., Viljanen, A., Usoskin, I., Vainio, R., Mirme, S., Mirme, A., Minikin, A., Petzold, A., Hõrrak, U., Plaß-Dülmer, C., Birmili, W., and Kerminen, V.-M.: Atmospheric data over a solar cycle: no connection between galactic cosmic rays and new particle formation, Atmos. Chem. Phys., 10, 1885–1898, https://doi.org/10.5194/acp-10-1885-2010, 2010."*

8. Line 82-83, the following paragraph: The aims of this paper are a bit unclear and phrased in a vague manner. I would suggest using a more precise phrasing. It could be explained what is meant by behaviour of charged particles. In addition, more details (i.e., formation and growth rates of charged particles are investigated) on what is actually done in the paper should be added.

Agreed. This paragraph now starts:

*"Here, the daily cycles, sources, and sinks of charged particles, as well as their contributions to new particle formation and growth rates were investigated in a summertime urban environment"*

9. Line 90-92: see my comments for the abstract. The classification used by Tammet (2006) was based on mass diameters, not mobility as is used here.

This has been amended, see above response.

**Materials and methods**

10. Line 100: please also state at what height from the ground the measurements are taken from.

We include the following line:

*"The charged and neutral particle measurements were taken from a laboratory on the fourth floor of an institute building positioned centrally within the Science Park, approximately 10 meters from ground level"*

11. Line 101: Potentially inaccurate phrasing. What does '… in excess of 100 nm from a number of highly-trafficked roads …' mean?

This now reads as follows:

*"Leipzig–TROPOS is located **approximately** 100 m from a number of highly–trafficked roads and is classified as an urban background site"*

12. Line 123: The NAIS measures air ion and total (neutral+charged), not neutral, particle number size distributions. In addition, the total particle concentrations are measured based on both the negative and positive polarity columns of the instrument. It could be mentioned, which data is used for the total particle concentration.

This now reads as follows:

*"A NAIS was used to measure the charged particle number size distribution (PNSD) from 0.8–42 nm, and the **neutral and charged** PNSD from 3–42 nm by their mobilities (3.2 to 0.0013 $cm^2\ V^{-1}\ s^{-1}$). **Neutral and charged measurements will hereon be referred to as simply "total", and the total measurements were taken from the negative column.**"*

13. Line 168-169: is the classification done based on the total PNSD or charged PNSD? This should be specified.

We used both the ion and particle size distributions. This is specified in the following lines

*"Each plot contained data spanning 24 hours and ranging from 0.8–42 nm (charged species from the*
*NAIS) and 3–800 nm (neutral particles from the NAIS and custom–built MPSS, combined). **All NPF***
***signatures were seen simultaneously in the PNSD and charged PNSD simultaneously**."*

14. Line 170: "… neutral particles ..." should read total.

Here we are referring to the neutral PNSD, not the total counts. We amend this to total PNSD.

15. Line 171: It is not clear what combined means. I am assuming the particle number size
distributions were combined using data below some diameter from NAIS and above it from MPSS.
Please clarify, and specify the connecting diameter.

Yes. We amend with the following line

"Total PNSD from the NAIS and custom–built MPSS, utilising the NAIS <20 nm and the MPSS >20
nm"

16. Line 171-172: Details of these plots such as color scale used are presented, yet none of these plots
are shown anywhere. I would either suggest removing the last sentence as it is not necessary, or
including contour plots in the analysis for added depth to the analysis.

We include a contour plot as an example in the supplement. We argue including notes about how NPF
events were identified down to the plotting of the data is important, as a bad colour palette (such as
base R's rainbow() palette can lead to misattribution of NPF events.

[Figure]

*Figure S1: Example contour plot. This is the hourly average mean contour plot for the entire campaign. Panel (a) shows the total (and charged) PNSD, while (b) shows the negative PNSD. The dashed lines show the upper cut of the charged measurements (A) and the lower cut of the total measurements (B).*

17. Line 174 and the following paragraph: it should be mentioned from which data CS is calculated from. I am assuming from the MPSS data.

Yes, this is from the MPSS data. We now state this in the text.

18. Line 186-187: Misleading phrasing "When calculating the formation rate …". This sentence makes it seems like the formation rate is the formation rate of particles with sizes in the size range, i.e., formation rate of 3 to 7.5 nm particles.

Correct, we have amended this.

*"When calculating the formation rate, instead of using a single particle size, a range is used. In this paper we use two ranges, 3–7.5 nm **for 3 nm particles**, and 7.5–22 nm **for 7.5 nm particles**. **These sizes were chosen** for consistency with the size–cuts used for the rest of the analyses"*

19. Line 193, Line 194: As aforementioned, I do not believe "charged species" is an entirely correct phrase to use in this context. Charged particles or ions would be better.

Amended to "charged particles" here and throughout.

20. There is absolutely no mention of black carbon (BC) anywhere in the methods section. No
mention of such data being used, or how it was measured. This information should be added to this
section.

Thanks for pointing this out. We include the following sentence

*"Black Carbon (BC) was measured through the attenuation of 880 nm light with an Aethalometer*
*(AE33, Magee Scientific, USA) using the default mass absorption coefficient."*

**Results and discussion**

21. Line 200: Table reference showing an error. There are some figure references later in the
manuscript, which are faulty too. Luckily, I was able to figure out what the tables and figures referred
to were.

Apologies. These have been amended.

22. Line 200, Line 202: Small ions can also include large charged molecules (see my comment for the
abstract section).

We have included a justification for our use of *particles* (see above).

23. Line 210-212: Some more recent studies could be referenced here too.

We have included a more recent reference here (see response to point 24).

24. Line 212-214: The electrode effect depends on the heights and is strongest near ground. As the
measurements are taken from the fourth floor, this should be addressed before making any
conclusions on the disparity of positive and negative small ion concentrations.

This is true! However, we do not believe there is any other reason for this disparity, except the
possibility of the walls of inlet influencing our measurements. We nonetheless comment on it.

*"The imbalance is believed to be caused by the Earth's negatively charged surface impacting the*
*distribution of charged species, referred to as the electrode effect (Hoppel, 1967; Hõrrak et al.,*
*2003). This effect is closest to the ground, and tapers off strongly at a height of meters (Hõrrak et al.,*
*2003). This may also be due to a charged surface on the wall near the inlet, or the inlet itself."*

25. Figure 2: are these the mean size distributions of charged particles? This should be stated both in
the figure caption, and in the text while referring to this figure for the first time.

Thank you, we now specify that this is a mean.

26. Line 224: This sentence "However, they were present in substantially larger concentrations …"
seems unnecessary and separate at this point, as the differences between NPF and non-event days are
not discussed yet. I would suggest leaving it out.

Good point. We have amended accordingly.

27. Lime 227-231: The size classification diameter limits in these studies are not exactly the same.
For small ions, Dos Santos et al. use 0.8-2 nm (in mobility diameters) while Tammet et al. use <1.6
nm (in mass diameters), while this study uses 0.8-1.6 nm (in mobility diameters). To some extent, this
can have an effect on the ion concentrations, especially as the sub-2 nm ion concentrations are
typically higher than above 2 nm ion concentrations.

This has been amended, see above response.

28. Line 231: Poor choice of words. "Observed variability …" indicates more to something observed
within this study, not to the differences between different studies. Perhaps "The differences between
these studies ..." would work better.

This now reads as follows

*"**The differences between these studies** may be explained by proximity to and density of the*
*surrounding transport infrastructure (see section 3.2.), photochemical processes (see section 3.5.),*
*and length of campaign period."*

29. Line 236-238: see my 26. comment.

This sentence has been deleted.

30. Line 238-243: Intermediate ion diameter range used in this study is from 1.6 to 7.5 nm and large
ion range is from 7.5-22 nm. The large ion diameter range is wider by over 8 nanometers. I do not see
how the comparison of the concentrations in these two size classes of very different widths is
meaningful. Considering this, attributing the differences in the concentrations of large and
intermediate ions to impact of air pollution does not seem justified if no other argument is given than
the concentrations of large ions being higher.

As these size classifications are often used in ions papers, we argue comparative concentrations are
useful. We do agree, however, that inferring too much from their ratios is not informative, and we
remove the final sentence of this paragraph.

31. Section 3.1 in general, Table 1: In addition to the mean values, median values and 5-95% spread
of the charged particle concentrations is given, yet these are not discussed anywhere. Looking at
them, we can for example notice that the 5% value of positive intermediate ions is larger than for
negative intermediate ions. However, 95% value of negative intermediate ions is larger than for
positive intermediate ions, Discussing the values aside from the mean concentrations would add depth
to the analysis.

We agree, and have included the following text (new text **bold**)

*"The positive particle concentrations are roughly a factor of 3 greater than the negative ion*
*concentrations, **and this is consistent across the 5-95% spread, so is not attributable to spikes in***
***positive charged particles**"*

*"Mean number concentrations of intermediate charged particles were 30.7 and 40.9 $cm^{-3}$ for positive*
*and negative polarities, respectively. **Negative particles show greater spread, with the lower 5% and***
***lower mean counts possibly also attributable to the electrode effect.**"*

*"Mean number concentrations were 210 and 205 $cm^{-3}$ for positive and negative polarities,*
*respectively, and were approximately 5-6 times higher (depending on polarity, higher for positive*
*particles) than intermediate charged particles. **The spread in large ion counts is similar between***
***positive and negative charged particles, and the relative magnitude of this spread is similar to the***
***intermediate ions**."*

32. Line 254-257: "Diurnal cycles suggest … ". I do not follow the reasoning here.

We include an extra reference to help argue our point as follows

*"Cosmic ray intensity is fairly constant throughout the lower atmosphere (Mercer and Wilson, 1965),*
*while the variations in radon concentrations is attributable to boundary layer dynamics (Čeliković et*
*al., 2023). The diurnal variation which we observe is therefore likely to be a combination of boundary*
*layer height changes affecting the radon concentrations, and variations in particle number surface*

*area altering coagulation rates due to both boundary layer height changes and primary and*
*secondary particle emissions."*

33. Line 262: Please specify what time midday corresponds to.

We now specify. This is 10:00 for intermediate, and 12:00 for large particles.

34. Line 276-282: I suggest including this part in the methods section instead. Also, it is still unclear
whether charged particle or total particle concentrations, or both, were considered when identifying
NPF events.

Great suggestion. We have moved this to the methods.

35. Line 287: Faulty reference again.

Apologies, this has been amended:

36. Line 288: Unclear phrasing, I suggest "... variables and concentrations of charged particles in
different mobility classifications ..." or similar.

Done. This now says:

*"Figure 3 shows the correlation coefficients between charged particles in different mobility*
*classifications and meteorological variables at Leipzig–TROPOS."*

37. Line 292-294, the following paragraph: "These trends align with expectations …". I believe a
reference should be added here. Rest of the discussion, i.e., the sentence starting "The parameter is
habitually related ...", in this paragraph could use some references too.

We have added the following reference to both of these sections

*"Air temperature is typically elevated when solar radiation is high, and relative humidity is typically*
*inversely related with air temperature (Seinfeld and Pandis 2016)"*

*"The parameter is related to air temperature, with cooler morning temperatures theoretically limiting*
*vertical mixing (Seinfeld and Pandis, 2016) and inadvertently enhancing small charged particle*
*concentrations."*

*"Seinfeld, J.H. and Pandis, S.N. (2016) Atmospheric Chemistry and Physics: From Air Pollution to*
*Climate Change. John Wiley & Sons, Hoboken."*

38. Line 302: I have doubts about photoionization having a significant contribution to intermediate or
large ion concentrations. Previous studies suggest that in the lower troposphere photoionization
should not have a significant impact on the ionization rates (see e.g., Harrison and Carslaw (2003)
https://doi.org/10.1029/2002RG000114), which is also stated in the references study by Jiang et al.
(2018). In addition, if photoionization contributed to ion concentrations, it should also do so for small
ions. I would argue that the observed correlation of solar radiation with intermediate and large ion
concentrations is attributable to photochemistry and NPF.

39. Figure 5 and the discussion starting from Line 312: I find the connection of Figure 5 and the
discussion in this paragraph with air ions unclear. The role of the discussion here for the manuscript
and its aims should be clarified.

Response to points 38 and 39: Great points, thank you. We provide the following argument now in the
text, which is more concise. We have also moved the figure with the meteorological data to the
supplement.

*"Concentrations of other acids ($HIO_3$, MSA) are an order of magnitude lower than $H_2SO_4$*

*concentrations, and so $H_2SO_4$ is the most likely candidate for the driver of NPF in this area.*

*Temperatures were high (~30 °C) during the campaign, and it is unlikely that OOMs can drive*

*particle formation in this data (Simon et al., 2020). The correlation between $H_2SO_4$ dimer and*

*charged particle concentration (**Figure 5, Figure S2**) shows that there is no statistically significant*

*correlation between $H_2SO_4$ dimer and small charged particles is, while the correlation with*

*intermediate and large ions is statistically significant. The correlation is strongest for the*

*intermediate ions, which peak coincidentally with $H_2SO_4$ dimer, which is coincident with high solar*

*radiation (**Figure 3**, **Figure S3**). Particle formation is accelerated by ionising radiation (Kirkby et*

*al., 2011; Kirkby et al., 2023), and a fraction of these new particles will be charged or will pick up*

*charge as they grow. NPF occurred on days with higher temperatures and solar radiation (**Figure***

***S3**) which is typical for ground-level NPF (Kerminen et al., 2018; Lee et al., 2019). High*

*temperatures can increase cluster evaporation rates, but this can be offset by the presence of ions*

*(Lee et al., 2019) although this is dependent on cluster composition (Kirkby et al., 2023). We attribute*

*these midday peaks in intermediate and large ions to NPF which is likely driven by sulfuric acid, and*

*argue that NPF is the major source of charged particles in this campaign (**Figure 2b**, **Figure S3**).*

*Primary emissions of intermediate and charged ions will be coincident with BC emissions (Thomas et*

*al., 2024)*

*Undefined and non-NPF events are observed when $H_2SO_4$ dimer is low. Undefined events are seen*

*when CS is high, and BC is higher than NPF event days, likely due to traffic emissions, and non-event*

*days occur when BC and CS are lower. Non-NPF days are possibly observed on these days due to low*

*concentrations of precursors. The morning and evening peaks in intermediate and large ions are*

*coincident with peaks in BC concentrations, and are therefore explicable by primary traffic emissions*

*(Thomas et al., 2024), and we argue that primary emissions are the second largest source of*

*intermediate and large ions in our data."*

And include the following new figures

[Figure]

*Figure 5: Correlation of H$_2$SO$_4$ dimer with small, intermediate, and large ions, coloured by date*

[Figure]

*Figure S2: Scatterplots (bottom panels), and histograms (upper diagonal) of meteorological*
*variables (solar radiation, air temperature, relative humidity, and wind speed) and small,*
*intermediate, large, and total charged particles (of both polarities). Also include are H₂SO₄ dimer*
*and BC. Red points are NPF days, green points are undefined days, and blue points are non-NPF*
*days.*

40. Line 344-346: Are there any potential explanations for the observation of lower small ion
concentrations on NPF event days?

No, we don't have the data to explain this, but posit that it may be due to stronger vertical mixing and
a deeper boundary layer on these days in the following line.

*"Small charged particle concentrations were lower on NPF event days compared to non–NPF event*
*days, consistent with findings in rural areas (Gagné et al., 2010; Hõrrak et al., 2003), possibly due to*
*stronger vertical mixing and a deeper boundary layer"*

41. Line 361: Is it a coincidence that BC concentrations were higher on nighttime on days, which NPF
event occurred compared to non-event days, or are there some potential explanations for it?

This could be because NPF days are coincident with clear skies and a shallow nocturnal boundary
layer.

*"BC peaks in the evening-time, possibly due to a shallow nocturnal boundary layer on these days."*

42. Line 376: "Observed similarities … " I found it difficult to understand what this sentence means.

We have rewritten this section as follows.

*"We attribute these midday peaks in intermediate and large ions to NPF which is likely driven by*
*sulfuric acid, and argue that NPF is the major source of charged particles in this campaign (Figure*
*2b, Figure S3). Primary emissions of intermediate and charged ions will be coincident with BC*
*emissions (Thomas et al., 2024)"*

*"The morning and evening peaks in intermediate and large ions are coincident with peaks in BC*
*concentrations, and are therefore explicable by primary traffic emissions (Thomas et al., 2024), and*
*we argue that primary emissions are the second largest source of intermediate and large ions in our*
*data."*

43. Line 387: It is not clear what "combined" means here. How I understood it is that the formation
rate of charged particles is determined as a sum of the formation rates of negative and positive ions.
Please clarify.

Yes, this is correct. This now reads

**"sum of both negative and positive particle formation rates; $J_{3-7.5}^{charged}$ and $J_{7.5-22}^{charged}$"**

44. Line 388: What does "combined" mean this context? Does this imply that the formation rate is
just the formation rate determined based on the total particle number size distributions, which are
measured by the NAIS. If so, the use of "combined" is unnecessary and confusing.

Yes, this is correct. In this instance, we remove "combined".

45. Line 389: This is a very interesting observation, which it implies that more particles are forming at
larger diameters than smaller diameters and that the survival probability of growing particles appears
to be over 1. Therefore, if accurate, something else aside from NPF such as traffic has a significant
contribution on the observed formation rate values. Some discussion on this and what are its
implications for the results of this study, such as on the contribution of ion-induced nucleation on
NPF, should be included.

We agree that this is interesting! It's also reflected in the shape of the charged PNSD (Figure 2). The
survival probability of new particles cannot, of course, be >1. We update the numbers in the
manuscript, as we were quoting J averaged across the whole campaign, which overemphasises
primary emissions. We instead now quote numbers just from NPF periods, which makes more sense,
and include the following discussion, alongside an updated figure that includes the diurnal cycle of Js:

*"Notably, the apparent J values of charged particles increased with aerosol size. The mean J values*
*of 3 and 7.5 nm charged particles during NPF were 0.165 and 0.326 cm$^{-3}$ s$^{-1}$, respectively, with mean*
*values of $J_{7.5-22}^{charged}$ approximately 2 times higher than $J_{3-7.5}^{charged}$. These compare with mean J values*
*of 3 and 7.5 nm total particles during NPF of 7.21 and 1.47 cm$^{-3}$ s$^{-1}$, respectively, with mean values of*
*$J_{7.5-22}^{total}$ approximately 0.20 times than $J_{3-7.5}^{total}$. The aforementioned J values are within the observed*
*tropospheric ranges for charged and total particles reported by Hirsikko et al. (2011). When*
*considering the calculated ratios of $J^{charged}$ / $J^{total}$ in the respective size ranges, the apparent mean*

*contributions of charged particles to 3 and 7.5 nm total particle formation were 5.7 and 12.7%,*
*respectively. $J_{3\text{-}7.5}^{total}$ is higher than $J_{7.5\text{-}22}^{total}$, which is typical, as new particles are lost as they grow*
*from 3 to 7.5 nm. However, $J_{3\text{-}7.5}^{charged}$ is higher than $J_{7.5\text{-}22}^{charged}$. We attribute this to charging of*
*growing aerosol by the condensation of smaller charged particles, and this is consistent with the low*
*concentrations of intermediate charged particles (Figure 2, Table 1). The diurnal cycle in J shows a*
*peak that is coincident with the peaks in $H_2SO_4$ dimer and intermediate charged ion concentrations*
*(Figure 5)."*

[Figure]

*Figure 6: Apparent formation rates of (A) 3–7.5 nm charged particles (left) and total particles (right) and (B) 7.5–22 nm charged particles (left) and total particles (right). Calculated from 9 new particle formation (NPF) event days using 10–minute means. (C) the diurnal cycle in formation rates on NPF days, and (D) growth rates (GR) of 3–7.5 and 7.5–22 nm charged particles. The coloured rectangle represents the middle 50% of the data, with the central horizontal line indicating the median value. The whiskers (vertical lines) extending from the rectangle show the spread of the data. Data points beyond the whiskers show outliers.*

46. Line 396-405: Some references to previous studies would be appreciated. ; 47. Line 405-406: "It
would be reasonable to view …" I do not understand/follow the reasoning here. Please clarify.

We have rewritten this for clarity. We do not include a reference here as it's a general statement
about surface area and mobility.

*"Large charged particles are more likely to act as a sink because of their greater surface area. In*
*comparison, smaller charged particles are more susceptible to ion–ion recombination due to higher*
*mobility. This recombination process, wherein two oppositely charged particles combine and*
*neutralise each other, accounted for in equation (4), can impact the abundance of smaller charged*
*species, influencing their ability to contribute to nucleation and particle formation in the atmosphere.*
*It would be reasonable to view $J_{3-7.5}^{charged}$ as an upper limit to ion–induced nucleation, while larger*
*charged particles appear to have a substantial contribution from charges acquired subsequently."*

48. Section 3.6 in general: I would suggest also including the formation rates of negative and positive
ions separately (and not just the combined value) in the analysis/discussion.

These are similar, and we include the following line in the discussion.

*"The ratio of $J^{positive}:J^{negative}$ is 0.9."*

49. Line 421: It should be clarified how the GRs of charged particles have been determined. As only
one GR per size range is presented, I would assume that the number size distributions of negative and
positive ions have been summed and from those a single GR value was derived. Similarly to Section
3.6, I suggest also including GRs of positive and negative ions in the analysis/discussion separately.

This is correct. The time evolution of the PNSD on NPF/non-NPF days is similar, so we opt to not re-
calculate these individually, but they were performed on the negative ion distributions. We include the
following lines in the manuscript

*"Growth rates were calculated according to the mode-fitting method outlined in Kulmala et al.*
*(2012)."*

50. Line 427-428: "Contrary to …". A reference is needed here.

We have included an appropriate reference

51. Figure 3, 5, 6, 7: Please specify also that the lines are mean number concentrations for each hour.

We have added this to every figure caption.

**Conclusions**

52. Line 443: it could be stated here in the beginning of the conclusions what diameter ranges small,
intermediate and large ions correspond to.

This has been added.

53. Line 442: I still do not understand this direct comparison of the concentrations in the different size
classifications as the diameter range widths are completely different. I do not find the observation of
large ions (7.5-22 nm) and small ions (0.8-1.6 nm) having similar concentrations meaningful as the
former covers so much larger range of ion sizes compared to the latter.

We agree and have removed this observation.

54. Line 443: Unclear phrasing "Variable concentrations were observed …" Variable concentrations
as compared to what? Additionally, observed suggest that something is observed in this study. A

better phrasing would be "The concentrations of intermediate/large ions in this study were observed to
be lower/higher than in some previously published studies, possibly linked to .."

For clarity we exclude this statement.

55. Please mention the measurement period in the conclusions section, for example in the beginning
of the section. In addition, mention at least that a NAIS was used to measure the charged particle/air
ion concentrations.

We agree, and now start the conclusions

*"The charged and total PNSDs were measured from 27ᵗʰ July to 25ᵗʰ August 2022 using NAIS in*
*urban Leipzig to understand the sources, sinks, and dynamics of charged particles.* Throughout the
measurement campaign, small (0.8–1.6 nm)…"

**Technical comments**

Line 54: … in the atmosphere, which … ; Line 55: missing word. These *ions* can be … ; Line 122: It
should read "a NAIS". ; Line 171: Missing word after "Each". ; Line 178: "assumed *to be* sulphuric
acid" ; Section 2.4: the symbols denoting parameters, such as D, β, etc should be in italics in the text.

Thanks, we have implemented all of these.

Line 241, 347, 465: "depending on polarity" does not clarify which value corresponds to which -
polarity.

Thanks, we have amended this throughout.

Line 376: suggest replacing "source strengths" with "concentrations".

You are right that concentrations are a mix of source and sink, both of which are important. This has
been amended

Line 473: A missing word.

We are not sure a word is missing here

Table legends and figure legends (Line 723-): The table and figure numbers are wrong.

Amended.

**Reviewer: 3**

The manuscript by Rowell et al. studied the role of air ions during atmospheric new particle formation
in urban Leipzig based on data collected from a one-month campaign. The authors investigated the
features in air ions in relation to selected meteorological parameters, CS, BC and H2SO4 dimer on
NPF days compared with those on non-NPF and undefined days as well as characterised their
formation rates and growth rates. Although the work is based only on a short campaign, it is a
valuable dataset worth publication contributing to the urban studies. However, the current manuscript
has several defects that cause confusion regarding especially size range classification and neutral
fraction definition. Also the one-month dataset cannot support the conclusion that ' ion–induced
processes play a minor role compared to neutral species in NPF at Leipzig–TROPOS'. Such general
conclusion requires long-term measurements. I would also like to suggest that the authors take a
closer look at the CI data, which could possibly help the elucidation of the precursor differences
between undefined and non-NPNF days. Adding further discussion on the impact of urban pollution
on NPF at the site will make the manuscript more valuable.

We thank the reviewer for their comments and agree they have strengthened the manuscript. Answers
below.

1. First of all, the authors stated that 'the air ion/charged particle population was mobility
classified …' but then gave size ranges in nanometers. It is confusing. Also size classification
in Tammet (2006) is based on mass diameter. NAIS measures mobility diameter in the range
of 0.8-42 nm. The authors stated that they followed the classification used by Tammet (2006).
A mobility diameter of 0.8 nm is around 0.4 nm in mass diameter. So have the authors
omitted the smallest ions?

We agree these are erroneous. For consistency with BSMA measurements, we have converted all
sizecuts to the appropriate sizecuts following the suggestion of Ku & de la Mora (2009), and making
the alteration using the effective gas diameter of 0.3 nm. We have reproduced all figures and
reworked all relevant parts of the text (not included below for length considerations).

Ku, B. K., & de la Mora, J. F. (2009). Relation between Electrical Mobility, Mass, and Size for
Nanodrops 1–6.5 nm in Diameter in Air. *Aerosol Science and Technology*, *43*(3), 241–249.
https://doi.org/10.1080/02786820802590510

2. On P5 L123-124, the authors wrote 'neutral PNSD from 3–42 nm by their mobilities (3.2 to
0.0013 cm2 V-1 s-1)'. The mobility range and the size range don't match. A mobility of 3.2
cm2/Vs is approximately 0.8 nm in mobility diameter. Also the sentence is confusing. NAIS
measures in the mobility size range of 0.8-42 nm, which applies in both air ion and total
particle modes. However, since corona charging is used in the total particle mode, data below
approximately 2.5-3 nm are contaminated by the charger ions and therefore are not usable.

Great point, thank you for highlighting the error. This now reads as follows

*"A NAIS was used to measure the particle number size distribution (PNSD) of naturally charged, and*
*also the sum of naturally charged and neutral species from 0.8–42 nm (3.2 to 0.0013 cm$^2$ V$^{-1}$ s$^{-1}$) by*
*their mobilities. In the case of the charged and neutral species, the data from 3-42 nm is used, as the*
*charging mechanism for neutral particles causes interference <3 nm."*

3. On P6 L170, the authors stated that they used combined data from NAIS and MPSS to get
neutral particles in the range of 3-800 nm. How was the neutral fraction obtained?Is there an
ion filter in the MPSS?

Sorry, this is a misattribution. We should really say neutral and charged particles. This has been
amended (and we also note how we joined together the size distribution)

*"neutral and charged PNSD from the NAIS and custom–built MPSS, utilising the NAIS <20 nm and*
*the MPSS >20 nm"*

And lower down we say

*"Neutral and charged measurements will hereon be referred to as simply "total""*

4. P7 L202: 'small charged particles (0.8-1.6 nm)', these are rather clusters.

We agree that these are clusters, however, we opt to use particles as a catch-all term (in the way that
*particle* encompasses everything from a large aerosol to a subatomic particle.) this way, we include
any potential for measurement of charged atoms, molecules, clusters of molecules, or charged
aerosols. The more commonly used *air ion* seems like a misattribution, as the term *ion* typically refers
to single atoms or molecules, but not larger particles. We explain this in the following sentence

*"Here, we refer to all charged species measured by the NAIS as "charged particles", which includes*
*charged aerosols, as well as charged molecules and charged clusters of molecules."*

5. P8 L214: the earth electrode effect is typically only pronounced at ground surface level. The data in this study was obtained from 4th floor. At this height, the earth electric field effect is small. The building's wall may have an influence. How was the NAIS inlet constructed? The high mobility channel of NAIS may also suffer from electric noise. Are the concentrations comparable between polarities in indoor environment?

Great point, thanks. We measured with conductive flexible rubber tubing, but the inlet was close to the wall of the building. We amend this as follows

*"The imbalance is believed to be caused by the Earth's negatively charged surface impacting the distribution of charged species, referred to as the electrode effect (Hoppel, 1967; Hõrrak et al., 2003). This effect is closest to the ground, and tapers off strongly at a height of meters (Hõrrak et al., 2003). This may also be due to a charged surface on the wall near the inlet, or the inlet itself."*

6. L283-285: NPF days have strong seasonal dependence. It is better to make comparison with studies in summer from other sites.

Sorry, we should specify, that is the summertime frequency from Bousiotis et al. This now says:

*"The frequency of NPF event days (30%) was comparable with frequencies from long–term analysis of summertime data at this site (Bousiotis et al., 2021)."*

7. L321-322: 'charged particles may play a significant role in stabilising clusters'. It is confusing that particles could stabilise clusters. Please change charged particles to charges.

In line with the suggestion from another reviewer, we in fact rewrite this section and so exclude this sentence altogether. The relevant section is included in our response to your point 9.

8. L374-375: Fig. 7 shows that BC on non-event days is comparable to that on undefined days.

We now clarify this as follows

*"Undefined events are seen when CS is high, and BC is higher than NPF event days, likely due to traffic emissions"*

9. L375-376: ' Non-NPF days are possibly observed on these days due to low source strengths of precursors.' The authors have access to the CI data which should be able to provide more details.

Yes. In an effort to amend this, as well as a couple of other comments on this section, this now reads as follows:

[revised manuscript text omitted]

10. L387: '…and neutral particles (charged and neutral particles, combined; J3–7.5neutral and J7.5–22neutral) '. The authors wrote 'neutral particles' but in the bracket 'charged and neutral particles, combined' . Are they charged or not? Or total particles?

This was to provide clarity. However, we now amend all use of "neutral" in this context to "total". Also, as we already discuss this in the methods, we remove these words.

11. Also on P19, the authors sometimes discussed about total particles and sometimes neutral particles. Very confusing.

This section has been rewritten and now uses consistent terminology

*"**Figure 6a,b** shows the apparent formation rates (J) of 3 and 7.5 nm charged particles (sum of both*
*negative and positive particle formation rates; $J_{3-7.5}^{charged}$ and $J_{7.5-22}^{charged}$) and total particles ($J_{3-7.5}^{total}$*
*and $J_{7.5-22}^{total}$) during NPF event days at Leipzig–TROPOS. **Figure 6c** shows the diurnal cycle of*
*these rates. The ratio of $J^{positive}:J^{negative}$ is 0.9. Notably, the apparent J values of charged particles*
*increased with aerosol size. The mean J values of 3 and 7.5 nm charged particles during NPF were*
*0.165 and 0.326 $cm^{-3} s^{-1}$, respectively, with mean values of $J_{7.5-22}^{charged}$ approximately 2 times higher*
*than $J_{3-7.5}^{charged}$. These compare with mean J values of 3 and 7.5 nm total particles during NPF of 7.21*
*and 1.47 $cm^{-3} s^{-1}$, respectively, with mean values of $J_{7.5-22}^{total}$ approximately 0.68 times than $J_{3-7.5}^{total}$.*
*The aforementioned J values are within the observed tropospheric ranges for charged and total*
*particles reported by Hirsikko et al. (2011). When considering the calculated ratios of $J^{charged} / J^{total}$ in*
*the respective size ranges, the apparent mean contributions of charged particles to 3 and 7.5 nm total*
*particle formation were 5.7 and 12.7%, respectively. $J_{3-7.5}^{total}$ is higher than $J_{7.5-22}^{total}$, which is typical,*
*as new particles are lost as they grow from 3 to 7.5 nm. However, $J_{3-7.5}^{charged}$ is higher than $J_{7.5-}$*
*$_{22}^{charged}$."*

12. The study is based only on a one-month campaign. It is not evident enough to reach the
conclusion that 'ion–induced processes play a minor role compared to neutral species in NPF
at Leipzig–TROPOS'. The generalisation requires studies from long-term measurement.

We provide the following alteration to this statement

*"Nevertheless, observed ratios of charged to uncharged particles in the size range impacted by NPF*
*suggest charged species play a minor role compared to neutral species in NPF at Leipzig–TROPOS*
*in our data"*

13. The reference list is messy. Please follow the alphabetic order and use the format of surname
followed by abbreviation of given name.

The reference list has been tidied, thank you.

Other issues:

L111: change 'city's weather' to 'the weather of the city' ;  L200, L208,L286, etc.: Error! Reference
source not found. Please check figures and tables. ; L200-201: 'large' is split. L287: 'variables' is
split.

Thank you for highlighting these, they have been amended.

---

## Author Response (AR2)

**The behaviour of charged particles (ions) during new particle formation events in urban Leipzig (Germany). Response to reviewers round 2.**

Note: Review comments are displayed in plain text, responses to those comments are displayed in blue and sections that have been added to the text are coloured *green (and italicised)* We thank the reviewers for their insightful comments and provide responses below.

**Reviewer: 2**

1. Fig. 5: the p-values are hard to read. Their location should be adjusted, or they could alternatively be mentioned in the figure caption

Thanks for the suggestion, we have opted to include them in the figure caption. Please see below in bold.

[Figure]

12

*Figure 5: Correlation of $H_2SO_4$ dimer with small, intermediate, and large ions, coloured by date.* **The $R^2$ values are 0.0014, 0.27, and 0.079, respectively, and the p values are >0.05, <0.05, and <0.05, respectively.**

2. Fig. 4: The lines in this figure may be hard to separate from each other for people with some form of colour blindness. I suggest going through all the figures with Coblis – Color Blindness Simulator

Great suggestion, thanks. We have fixed these figures and swapped the palette for a similar, but colourblind friendly one. Please see figure below.

[Figure]

21

*Figure 4: Mean diurnal cycles of (a) small (0.8–1.6 nm), (c) intermediate (1.6–7.5 nm), and (e) large*
*(7.5–22 nm) charged particles, as well as (b) BC, (d) $H_2SO_4$ dimer, and (f) CS on new particle formation*
*(NPF) event, undefined, and non–NPF event days. The vertical lines represent the standard error of*
*the mean.*

3.  Line 303: "…mobility classifications…". Should it not be diameter classifications?

Correct, thanks for highlighting this

4.  Fig. 6: In the title, it would be tidier if Jion → Jion

Following this and a suggestion from reviewer #3, we have tidied up the text on this figure. Please see
the below.

[Figure]

31

32  *"Figure 6: Apparent formation rates of (A) 3–7.5 nm charged particles (left) and total particles*
33  *(right) and (B) 7.5–22 nm charged particles (left) and total particles (right). Calculated from*
34  *9 new particle formation (NPF) event days using 10–minute means. (C) the diurnal cycle in*
35  *formation rates on NPF days, and (D) growth rates (GR) of 3–7.5 and 7.5–22 nm charged*
36  *particles. The coloured rectangle represents the middle 50% of the data, with the central*
37  *horizontal line indicating the median value. The whiskers (vertical lines) extending from the*
38  *rectangle show the spread of the data. Data points beyond the whiskers show outliers."*

39  5.  Line 215: the parentheses and comma from "(change of dp over time,)" should be removed.

40  Thanks, this has been done

41  6.  Line 412: "$J_{3-7.5}$charged is higher than $J_{7.5-22}$charged" should be $J_{3-7.5}$charged is lower
42      than $J_{7.5-22}$charged

43  This has been corrected.

**Reviewer: 3**

In the revised manuscript the authors stated that the charged particles were classified into small (0.8-1.6 nm), intermediate (1.7 – 7.5 nm) and large (7.5 – 22 nm) fractions by mass diameter. Yet the authors explained that their size conversion was based on Ku & de la Mora (2009). I am afraid it is getting more confusing. Ku & de la Mora (2009) provided a means to link mobility diameter to mass diameter. If what you get from the size conversion was dp as defined in Ku & de al Mora (2009), your size classification is actually based on mobility diameter. For example, a mobility of 3.2 $cm^2V^{-1}s^{-1}$ is about 0.8 in mobility diameter. For better clarification, please include the mobiulity range for your size classification.

Ku and de la Mora (2009) provide an approximation for the effective gas diameter, which can be used to convert mass diameter to mobility diameter (i.e., $d_e = d_m + D_g$ where $d_e$ is the electrical mobility diameter, $d_m$ is the mass diameter, and $D_g$ is the effective gas diameter, which is roughly .3 nm). We converted the size cuts from previous papers (for example, 0.8 – 1.6 nm mass diameter) to a mobility diameter and reperformed our analyses (1.1 – 1.9 nm). For clarity, we include the following

*"The air ion/charged particle population was classified into small (0.8–1.6 nm mass diameter, 1.1—1.9 nm electrical mobility diameter), intermediate (1.6–7.5 nm mass diameter, 1.9—7.8 nm electrical mobility diameter), and large particles (7.5–22 nm mass diameter, 7.8 – 22.3 nm electrical mobility diameter) for analysis, following the classification system outlined by Tammet (2006)."*

And later on…

*"A NAIS was used to measure the particle number size distribution (PNSD) of naturally charged, and also the sum of naturally charged and neutral particles from 0.8–42 nm (3.2 to 0.0013 $cm^2\ V^{-1}\ s^{-1}$) by their mobilities. From here onwards we refer to all diameters as mass diameters for consistency with the literature (e.g. Tammet et al., 2006; Ku & Fernandez de la Mora, 2009)."*

On P26L34: "Observed ratios of charged to uncharged particles…": shouldn't it be total? The same issue is also in the conclusions

Yes, thank you. We have fixed it in both locations.

Technical issues: the authors used charged and total in the superscript of formation rates and size ranges in the subscript. However, Fig. 6 was presented with a completely different notation scheme.

In line with this and a recommendation from reviewer #1, we have revised this figure and present it below

[Figure]

74

*"Figure 6: Apparent formation rates of (A) 3–7.5 nm charged particles (left) and total particles (right) and (B) 7.5–22 nm charged particles (left) and total particles (right). Calculated from 9 new particle formation (NPF) event days using 10–minute means. (C) the diurnal cycle in formation rates on NPF days, and (D) growth rates (GR) of 3–7.5 and 7.5–22 nm charged particles. The coloured rectangle represents the middle 50% of the data, with the central horizontal line indicating the median value. The whiskers (vertical lines) extending from the rectangle show the spread of the data. Data points beyond the whiskers show outliers."*